# Upwelled plankton community modulates surface bloom succession and nutrient availability in a natural plankton assemblage

Allanah J. Paul[1]*, Lennart T. Bach[2], Javier Arístegui[3], Elisabeth von der Esch[1], Nauzet Hernández-Hernández[3], Jonna Piiparinen[4], Laura Ramajo[6,7,8], Kristian Spilling[4,8], Ulf Riebesell[1]

[1] GEOMAR Helmholtz Centre for Ocean Research Kiel, Düsternbrooker Weg 20, 24105 Kiel, Germany

[2] Institute for Marine and Antarctic Studies, University of Tasmania, Hobart, Tasmania, Australia

[3] Instituto de Oceanografía y Cambio Global (IOCAG), Universidad de Las Palmas de Gran Canaria (ULPGC), Las Palmas, Spain

[4] Marine Research Centre, Finnish Environment Institute, Helsinki, Finland

[5] Center for Advanced Studies in Arid Zones (CEAZA), Coquimbo, Chile

[6] Departamento de Biología Marina, Facultad de Ciencias del Mar, Universidad Católica del Norte (UCN), Coquimbo, Chile

[7] Center for Climate and Resilience Research (CR)2, Santiago, Chile

[8] Centre for Coastal Research, University of Agder, Kristiansand, Norway

*Correspondence to:* Allanah J. Paul (apaul@geomar.de)

**Abstract.** Upwelling of nutrient rich waters into the sunlit surface layer of the ocean supports high primary productivity in Eastern Boundary Upwelling Systems (EBUS). However, subsurface waters not only contain macronutrients (N, P, Si) but also micronutrients, organic matter, and seed microbial communities that may modify the response to macronutrient inputs via upwelling. These additional factors are often neglected when investigating upwelling impacts on surface ocean productivity. Here, we investigated how different components of upwelled water (macronutrients, organic nutrients, seed communities) drive the response of surface plankton communities to upwelling in the Peruvian coastal zone. Results from our short term (10 days) study show that the most influential drivers in upwelled deep water are 1) the ratio of inorganic nutrients ($NO_x:PO_4^{3-}$) and 2) the microbial community present that can seed heterogeneity in phytoplankton succession and modify stoichiometry of residual inorganic nutrients after phytoplankton blooms. Hence, this study suggests that phytoplankton succession after upwelling is modified by factors other than the physical supply of inorganic nutrients. This would likely affect trophic transfer and overall productivity in these highly fertile marine ecosystems.

## 1 Introduction

The Humboldt Current System (HCS) in the South Pacific Ocean is considered the most productive upwelling region in terms of fish production and spans the coasts of northern Peru to Chile between 5 and ~45°S (Chavez and Messié, 2009). Alone the northern HCS off the Peruvian coast constitutes up to 20% of global industrial fish landings (Tarazona and Arntz, 2001) at a value of over $2bn USD to the Peruvian economy annually from exports in 2013 (Humala Tasso et al., 2015). This immense fish productivity is sustained by significant primary production, underpinned by wind driven upwelling along the continental shelf which occurs seasonally along the Chilean coast and almost permanently in the northern Humboldt Current off the Peruvian coast (Kämpf and Chapman, 2016). South easterly trade winds push surface waters offshore and westward towards the South Pacific subtropical gyre via Ekman transport. This movement induces upwelling of nutrient rich subsurface waters originating primarily in the poleward Peru–Chile Under Current (PCUC, Gutiérrez et al. (2016)) to the sunlit surface ocean, where primary producers assimilate the inorganic nutrients into organic matter.

However, the nutrient influx into the euphotic zone depends not only on the intensity of the wind-driven upwelling but also the characteristics of the source water present in the Ekman layer from which the upwelling occurs. For example, seasonal fluctuations in the strength of northward flowing Sub–Antarctic Water (SAW), and southward flowing Equatorial Sub–Surface

Water (ESSW) undercurrents can modify the source water oxygen and nutrient concentrations for coastal wind driven upwelling (Kämpf and Chapman, 2016). Changes in thermocline and nutricline depth due to interannual El Niño–Southern Oscillation (ENSO) phases has a similar impact on the upwelling source waters even if the upwelling depth does not change (Espinoza–Morriberón et al., 2017).

Another defining feature of HCS ecosystems is the extensive oxygen minimum zone (OMZ) in the eastern tropical south Pacific, extending up to 1000 km from the coast and over 600 m thick (Fuenzalida et al., 2009). These subsurface oxygen deficient waters are a result of a combination of three factors: oxygen–poor equatorial source water feeding the PCUC, slow ventilation, and high consumption of oxygen due to remineralisation of organic matter maintaining this oxygen deficiency (Pennington et al., 2006). Low oxygen concentrations facilitate significant loss of fixed nitrogen via anaerobic microbial
metabolism (annamox and denitrification, Lam et al. (2009)) and redox–dependent inputs of phosphate (P) and iron (Fe) from shelf sediments (Bruland et al., 2005). Thus, a biogeochemical imprint of nitrate (N) deficiency and an excess of P prevails in the inorganic nutrient stoichiometry of waters upwelled along the Peruvian shelf.

Cross–shelf shifts in the dominant primary producers have been linked to not only nutrient concentrations but also the relative proportion of N to P present (Franz et al., 2012; Meyer et al., 2017). Generally speaking, coastal phytoplankton communities
on the Peruvian shelf in the northern HCS are dominated by diatoms or dinoflagellates, rapidly growing phytoplankton groups which capitalise on the abundance of inorganic nutrients in the freshly upwelled water along the shelf. Although N:P ratios are low in the upwelled water, the high concentrations mean that these groups have the luxury of assimilating nutrients in close to Redfield proportions which meets their physiological requirements for growth and nutrient acquisition (Arrigo, 2005). In nutrient deplete water further offshore, smaller phytoplankton such as picocyanobacteria become more abundant (Franz et al.,
60   2012).

Indeed, blooms of different phytoplankton populations can easily be induced experimentally by the addition of inorganic nutrients with a different N:P (Czerny et al., 2016; Hauss et al., 2012). However, addition of inorganic nutrients to a surface community neglects the dilution of the surface community when deep water with lower phytoplankton abundances is mixed upon upwelling. This would drive a phytoplankton-dominant response and modify trophic relationships between consumers
and phytoplankton in a similar way to eutrophication studies (Taylor et al., 1995). Messie and Chavez (2015) suggest this dilution effect may underlie the Peruvian productivity paradox where highest upwelling is out of phase seasonally with the highest detected surface chlorophyll concentrations (Chavez and Messié, 2009). Furthermore, subsurface waters in the region have dissolved organic matter concentration and composition (Loginova et al., 2019), and trace metal concentrations that are different to those in surface waters and depends on the history of the water. For example, this can be influenced by the
predominance of heterotrophic organisms in aphotic subsurface waters (e.g. Schmidt et al., 2016) and contact of sub– or anoxic water with sediment on the seafloor (Bruland et al., 2005). This may also modify the response of the surface plankton assemblage.

While surface phytoplankton blooms in upwelling regions are stimulated mostly by the nutrients brought to the surface, the key phytoplankton group can be altered by organisms in the deep water also brought to the surface that can seed these blooms.
Recently sunk algal cells or dormant life stages of diatoms or dinoflagellates (Smayda and Trainer, 2010) may be present or reintroduced via resuspension of cells in surface sediments (Ishikawa and Furuya, 2004) to aphotic subsurface waters where upwelling occurs from. Once exposed to light in the photic zone and combined with the nutrient rich upwelled water, these resting algal cells or spores can germinate thereby inoculating a fresh bloom (Carreto et al., 2016). Horizontal mixing of surface waters following relaxation of vertical upwelling or along fronts can also introduce new phytoplankton populations that can
propagate blooms spatially and lead to a succession in the dominant phytoplankton groups (Smayda and Trainer, 2010).

A mesocosm study investigating the impacts of upwelling and nutrient stoichiometry on the northern Humboldt Current System pelagic ecosystem was carried out during 2017 (Bach et al. 2020). This study primarily looked at the ecosystem level response of a natural surface plankton community in terms of biogeochemistry and ecology to addition of subsurface waters, but cannot disentangle which property or properties (inorganic nutrient concentrations and stoichiometry, dissolved organic nutrient and
trace metal signature, subsurface plankton community) drives this response. Hence, we designed a complimentary experiment to investigate these three drivers on the lower food web response more in depth and in parallel to the mesocosm study. In particular we wanted to understand what impact does the subsurface-water chemistry (inorganic nutrients, organic nutrients)

and biology (seed populations) have on phytoplankton bloom biomass and phytoplankton community composition and succession and hence, what are the implications for nutrient turnover in the coastal Peruvian upwelling system.

## 2 Materials and methods

### 2.1 Experimental design

The experiment was setup with six treatment combinations to disentangle components of upwelled deep water that may influence surface phytoplankton blooms. Three deep water components were used to distinguish the impact of 1) inorganic macronutrient ratio between nitrate and phosphate ("inorganic"), 2) organic nutrients and other micronutrients such as trace
metals ("organic"), and 3) the seed microbial community ("biology"). We also selected two different subsurface water sources with two different nutrient levels (HN = high nitrate, LN = low nitrate). These two different N concentrations were selected to distinguish the impact of nitrate concentration and N:P stoichiometry on the plankton response patterns (Fig. 1). Treatment combinations after referred to as a combination of the nutrient level and the deep water component e.g. "HN inorganic" or "LN biology".


**Figure 1: Experimental design indicating the six treatment combinations implemented. 'Filtered' refers to filtration through 0.1 µm. Each of the six treatment combinations had four replicates. Sources of inorganic/organic nutrients and the microbial community are in addition to the 50% of mesocosm surface water used as a base in all six treatment combinations. Details of the water collection and treatment implementation are provided in the Sect. 2.2.**

<table>
<tr><th colspan="4">Deep water component</th><th rowspan="2">Property</th></tr>
<tr><th></th><th><i>Inorganic</i></th><th><i>Organic</i><br>(+ Inorganic)</th><th><i>Biology</i><br>(+ Inorganic<br>+ Organic)</th></tr>
<tr><td rowspan="6"><b>Nutrient level</b></td><td>--</td><td>Station A</td><td>Station A</td><td>Deep water station</td></tr>
<tr><td>filtered surface water</td><td>filtered deep water</td><td>deep water</td><td>Water type added (50%)</td></tr>
<tr><td>+ nitrate<br>+ phosphate</td><td>deep water</td><td>deep water</td><td>Inorganic nutrient source</td></tr>
<tr><td>surface water</td><td>deep water</td><td>deep water</td><td>Organic nutrient source</td></tr>
<tr><td>--</td><td>--</td><td>deep water</td><td>Microbial community source</td></tr>
<tr><td>●</td><td>●</td><td>●</td><td>Colour code</td></tr>
</table>

<i>High nitrate (HN)</i> spans the first six data rows above; <i>Low nitrate (LN)</i> spans the rows below.

<table>
<tr><td rowspan="4"></td><td>--</td><td>Station B</td><td>Station B</td><td>Deep water station</td></tr>
<tr><td>filtered surface water</td><td>filtered deep water</td><td>deep water</td><td>Water type added (50%)</td></tr>
<tr><td>+ nitrate<br>+ phosphate</td><td>deep water</td><td>deep water</td><td>Inorganic nutrient source</td></tr>
<tr><td>surface water</td><td>deep water</td><td>deep water</td><td>Organic nutrient source</td></tr>
</table>

| | | | |
|---|---|---|---|
| -- | -- | deep water | Microbial community source |
| ● (grey) | ● (blue) | ● (green) | Colour code |

## 2.2 Water collection, incubation set up and sampling

Subsurface water was collected on 16[th] March 2017 from two stations (Station A: 12°02,840 S; 077°17,064 W; Station B: 12°02,249 S; 77°40,123 W) on board R/V Humboldt (Fig. 1). These stations are part of the Linea Callao time series transect that is regularly sampled for inorganic nutrient concentrations and water column properties by Instituto del Mar del Perú (IMARPE). These data indicated that the offshore station (A) and more coastal station (B) usually have different nutrient profiles, in particular nitrate concentrations (see e.g. Graco et al., 2019).

To select the sampling depths, we performed CTD profiles using a CTD60M probe (Sea and Sun Technology) with dissolved oxygen ($O_2$) and hydrogen sulphide ($H_2S$) sensors at both stations down to a maximum depth of 150 m. $O_2$ and $H_2S$ concentrations were used to indicate oxygenated and non-sulfidic waters likely containing inorganic nitrogen. We then made multiple deployments of four Niskin bottles attached in series (depth range of 15 m) to collect a total of 100 L of seawater at station into acid-cleaned carboys (Fig. 1). We took subsamples directly from each carboy to determine the precise nutrient concentrations of the collected subsurface water. These samples were filtered (0.45 µm Sterivex, Merck Millipore) and stored in a cool box in the dark until analysis on the same day on shore (see Sect. 2.3 for details on the methods of nutrient analysis). Note, although this subsurface water was collected at relatively modest depths (>40 m), it is hereafter referred to as "deep water" to clearly distinguish it from surface waters collected from the mesocosms (M in Fig. 2).

One day later (17[th] March 2017, Day 20 of mesocosm study), 40 L of nutrient depleted surface water was collected from the photic, oxic layer in five of the eight mesocosms (Station Mesocosm (M); M1-5, see Bach et al. (2020) for more details) using a manual vacuum pump (pressure < 300 mbar). The water was pooled in clean carboys. The mesocosm plankton community was in a post-bloom phase where inorganic nitrogen was low and a sub-surface Chl *a* maximum had developed between 5-15 m depth (Bach et al., 2020). Precise sampling locations, depths and measured nutrients of collected water are summarised in Fig. 2. While neither trace metal nor dissolved organic nutrient measurements were made from this study to characterise the deep water sources, these were assumed to be different between Station A and B, just as the inorganic nutrient concentrations were.

**Figure 2: Map of sampling locations off the Peruvian coast indicating collection sites and characteristics of water used in the incubations.**

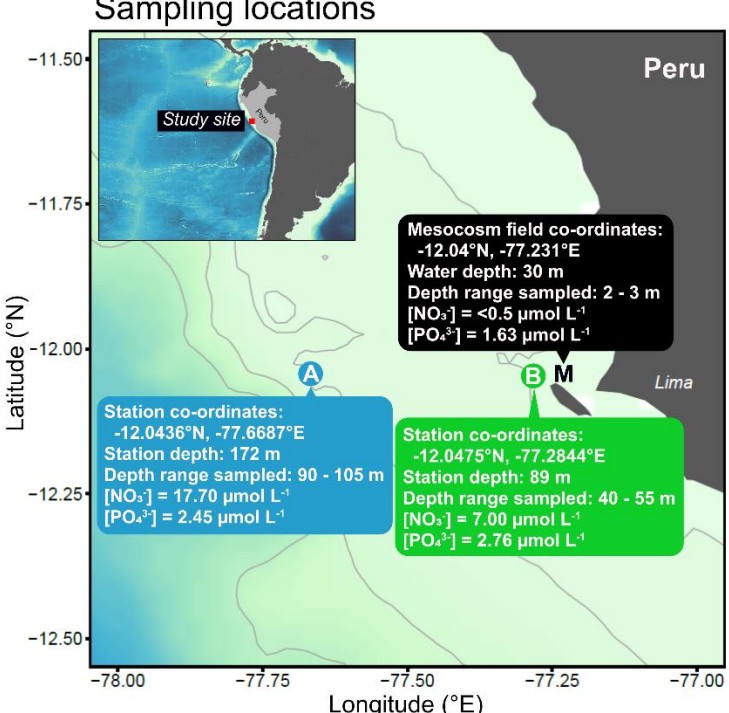

After collection, both the deep water and the surface (mesocosm) water were screened using a 64 µm gauze to remove large predators such as copepods and gelatinous organisms that can be patchily distributed and could exert unequal grazing pressure in these low volume (15 L) incubations. This screened water is hereafter referred to as "deep water" or "surface water", respectively. Filtered water, both surface and deep, was produced by stepwise filtration over gauze down to 10 µm mesh size, followed by sterile filtration using filter cartridges (0.1 µm, Whatman Polycap TC 36) to remove the microbial community while retaining the chemical properties (i.e. inorganic/organic nutrients). After the screening (<64 µm) and filtration (<0.1 µm) process, the treatment combination (HN inorganic, LN inorganic, HN organic, LN organic, HN biology, LN biology, Fig. 1) were prepared in six separate clean large plastic rainwater tanks (300 L volume). All six tanks received 100 L of surface water containing the mesocosm plankton community (<64 µm) and the ambient inorganic and organic nutrient concentrations.

For the HN and the LN "inorganic" tanks, this mesocosm surface water was mixed with 100 L of filtered (0.1 µm) mesocosm surface water. This surface water was nutrient deplete, hence, stock solutions of sodium nitrate (NaNO$_3$, EMPLURA®, Merck, Germany) and potassium dihydrogen phosphate (KH$_2$PO$_4$, EMSURE® ISO for analysis, Merck, Germany) dissolved in ultrapure water (MilliQ, Millipore) were added to mimic the same inorganic nitrate and phosphate concentrations as in the high nitrate (HN) and low nitrate (LN) deep water.

In the HN and LN "organic" tanks, this mesocosm surface water was mixed with 100 L of filtered deep water from either Station A (HN, high nitrate) or B (LN, low nitrate). In the HN and LN "biology" tanks, this mesocosm surface water was mixed with 100 L of deep water from either Station A or B. All tanks were carefully mixed before distributing the water among the four replicate flexible 15 L incubation containers per treatment. Directly after filling, samples for inorganic nutrient analysis and determination of phytoplankton abundances by flow cytometry were collected in duplicate from each replicate.

Once all 24 containers were filled, they were randomly placed in black incubator tubs covered with light foil with ~25% transmittance (Blue Lagoon, LEE filters). The incubators were situated outside in direct light and filled with natural seawater using a flow-through water system to maintain ambient seawater temperature. A submersible logger (HOBO pendant Temperature/Light Data Logger, Onset Computer Corporation, MA, U.S.A) measured mean temperature of 23.1 ± 3.5 °C (mean ± s.d.) over the 10-day study period. The high variability was due to temperature differences between day and night. The logger was shifted between incubators daily to measure conditions in all incubators. To calibrate the illuminance measured by the HOBO loggers to irradiance (PAR), we attached the logger to a CTD with a Photosynthetically Active Radiation (PAR) sensor (CTD60M probe, Sea and Sun Technology) and submerged this in ~1 m water depth between 11:50 and 17:00 on 14[th] April 2017. Using a calibration curve (Fig. S1), we estimated an average PAR of 250 µmol quanta m$^{-2}$ s$^{-1}$ during daylight hours.

Subsampling for inorganic nutrient concentrations and phytoplankton abundances were carried out daily and for all other variables (Chlorophyll $a$ (Chl $a$), enzyme activity, photophysiology and dissolved organic matter) approximately every 2[nd] day from the incubation containers (hereafter "incubations") starting at 08:30 for 10 days between the 19[th] and 28[th] March 2017. Sampling was rapid and took place on shore so all samples were taken to the laboratory within 1 hour and processed or analysed within 6 hours of sampling from the incubators.

## 2.3 Inorganic nutrient analyses

Inorganic nutrient concentrations were measured from one subsample per replicate. This water from each subsample was prefiltered (0.45 µm, Sterivex, Merck Millipore) into acid cleaned tubes before analysis on a continuous flow analyser (QuAAttro Autoanalyser, SEAL Analytical) using an autosampler (XY2 autosampler, SEAL Analytical) and a fluorescence detector (FP–2020, JASCO). Nitrate ($NO_3^-$) and nitrite ($NO_2^-$), hereafter reported as a combined $NO_x$, were determined colorimetrically according to Morris and Riley (1963), while silicate ($Si(OH)_4$ or DSi) and phosphate ($PO_4^{3-}$, also referred to as DIP) concentrations were determined colorimetrically according to Mullin and Riley (1955). Average limit of detections (LOD) were 0.123 µmol L$^{-1}$, 0.054 µmol L$^{-1}$, 0.033 µmol L$^{-1}$ and 0.336 µmol L$^{-1}$ for nitrate, nitrite, phosphate and silicate, respectively. Further details on measurements and their precision can be found in Bach et al. (2020).

Nutrient drawdown ratios for $NO_x$ and DIP ($\Delta NO_x$:$\Delta DIP$) were calculated on a daily basis according to Equation (1), where $[NO_x]_{t1}$ and $[DIP]_{t1}$ are the concentrations of $NO_x$ and DIP, respectively, on Day 1 and $t_n$ = Day n.

$$(\Delta NO_x : \Delta DIP)_{t_n} = \frac{[NO_x]_{t_1} - [NO_x]_{t_n}}{[DIP]_{t_1} - [DIP]_{t_n}} \qquad (1)$$

Dissolved silicate drawdown ($\Delta DSi$) for a given sampling day ($t_n$) was calculated in reference to initial concentrations measured on Day 1 according to Equation (2):

$$\Delta DSi = [DSi]_{t_1} - [DSi]_{t_n} \qquad (2)$$

## 2.4 Chlorophyll $a$ and phytoplankton community composition analyses

A subsample from each replicate container were taken on Day 1, then every 2[nd] day from Day 2 until Day 10 for the analysis of Chlorophyll $a$ (Chl $a$) concentrations. Volumes of between 200–400 mL were filtered onto glass fibre filters (GF/F, Ø 25 mm, nominal pore size 0.7µm, Whatman) with care taken to minimise exposure to light and maintain the vacuum pressure <200 mbar. Filters were stored at -20°C until extraction and analysis. Chl $a$ was extracted from the filters in 90% acetone in plastic vials using glass beads and a cell mill to burst the cells and release the pigments into the supernatant. Concentrations were measured in the supernatant according to Welschmeyer et al. (1994) on a Turner 10–AU fluorometer.

Seawater samples for phytoplankton community analysis were collected in 2 mL cryovials and measured on the same day without fixation on an BD Accuri$^{TM}$ C6 Flow cytometer. Samples were stored cooled in the dark until analysis within six hours of sampling. Each sample was analysed using fast flow rate (~66 µL per minute) to measure a total volume of 100 µL. No beads were added, instead particle sizes were determined via sequential size fractionations with polycarbonate filters of different pore size as described in Veldhuis and Kraay (2000). We used the excitation/emission wavelengths of FL3 = 488/670

nm for Chlorophyll *a*, FL2 = 488/585 nm for Phycoerythrin, and FL4 = 640/670 nm for Phycocyanin. Individual particles were gated into groups based on size-fractions (picoeukaryotes, nanophytoplankton, small microphytoplankton, larger microphytoplankton), taxonomic groups (*Synechococcus*, cryptophytes) and other forms (chains and group 'FL4') based on fluorescence signal and other properties such as size and shape using forward/side scatter (FSC/SSC) measurements. We considered all populations for the quantitative analysis. For gating, some identification was needed on specific fluorescence channels (e.g. *Synechococcus* on FL2) and these are then excluded from the other plot (e.g. FL3 vs. FSC) to avoid overlap with the other populations. Gating of the microphytoplankton groups based on size (small, large) was modified to the best fit for each sample, however, there is a source of uncertainty associated with this approach due to overlap in some samples between the groups (see Fig. S2 for two cytograms with identified groups). The "chain" group was distinguished by dividing the Chl *a* red fluorescence amplitude by Chl *a* red fluorescence height to class elongated cells, such as chain forming diatoms, by an elongated fluorescence signal. "FL4" were classed as very small particles that may be individual chloroplasts but could not be attributed to any likely phytoplankton group. An example cytogram to indicate the gating applied is provided in the Supplementary Material (Fig. S2). Contribution to fluorescence was calculated from the relative contributions of each gated group to total FL3 (Chl *a*) fluorescence (see Bach et al. (2019)).

### 2.5 Extracellular enzyme activity (leucine aminopeptidase)

The leucine aminopeptidase (LAP) activity was determined using a fluorometric assay with *L*-leucine 7-amido-4-methyl-coumarin (Leu-AMC; Sigma Aldrich) as a substrate (Stoecker and Gustafson, 2003). Leu-AMC was added to a final concentration of 500 µmol L$^{-1}$, a concentration which saturates the enzyme ($V_{max}$) according to separate preliminary kinetic tests. The samples (volume = 200 µL, except 400 µL on Day 1) were incubated in the dark at in situ surface temperature for a minimum of four hours. The fluorescence was measured every 30-60 min during the incubation period with a Cary Eclipse (Agilent Technologies) spectrofluorometer using 380 nm excitation and 440 nm emission wavelengths. The fluorescence emitted from the samples were compared with a standard curve determined using 7-amino-4-methyl-coumarin (AMC; Sigma Aldrich) dissolved in dimethyl sulfoxide (DMSO), and the LAP activity calculated by linear regression.

### 2.6 Fast Repetition Rate Fluorometry (FRRF) and Chromophoric Dissolved Organic Matter (CDOM) analyses

The samples for Fast Repetition Rate Fluorometry (FRRF) were collected in dark plastic bottles (125 mL) in order to avoid damage from light to the cells during the sampling period (< 1 h). To ensure dark adaptation of phytoplankton cells before analysis, samples were stored in the dark and at room temperature for at least 30 min right after they had arrived in the laboratory facilities of the Instituto del Mar del Perú (IMARPE). Thereafter, three subsamples plus a blank from each treatment were analysed by means of a Fast Induction and Relaxation (FIRe) technique and system (Satlantic FIRe System, for detailed information about FIRe see Gorbunov and Falkowski (2004)). The blanks were obtained by gravitational filtration of water samples through polycarbonate filters (PC, Ø 25 mm, pore size 0.2 µm, DHI) and subtraction of this measured baseline seawater signal from the corresponding sample signal. FIRe cuvettes were regularly cleaned (every 10–12 samples) with 5% HCl and gently rinsed with ultrapure (Milli-Q) water to avoid fouling. The maximum photochemical efficiencies ($F_v/F_m$) were estimated from FIRe profiles based on the biophysical model in Kolber et al. (1988).

Chromophoric Dissolved Organic Matter (CDOM) samples were collected in amber glass bottles (75 mL) to prevent potential photobleaching during sampling and transportation before measurement as described in Catalá et al. (2018). A modular spectrophotometer (Ocean Optics) consisting of a USB2000+UV−VIS ES detector connected via optical fibres to a DH2000BAC light source and to a 100 cm, 250 µL capillary (LPC100CM; World Precision Instruments; WPI), was used to measure CDOM absorption spectra from 200 to 900 nm at 1 nm intervals. Before analysis, the samples were gravitationally filtered through precombusted (5 h at 450ºC) glass fibre filters (GF/F, Ø 25mm, nominal pore size 0.7 µm, Whatman) and then run through the system at a constant rate of 1 mL min$^{-1}$. Constant flow rate was achieved by means of a peristaltic pump (ISMATEC). Ultrapure (Milli-Q, Millipore) water blanks were analysed after every sample. Absorption coefficients (m$^{-1}$) at 254 nm ($a_{254}$), 250 nm ($a_{250}$) and 365 nm ($a_{365}$) were calculated following Green and Blough (1994). $a_{254}$ has been used as DOC concentration proxy in the ocean (Catalá et al., 2018; Lønborg and Álvarez-Salgado, 2014), while the ratio of $a_{250}$ and $a_{365}$ ($E_2$:$E_3$) is commonly used as an indicator of DOM molecular weight (MW, Helms et al. (2008)).

## 2.7 Statistical analyses

All statistical tests were carried out in the R environment (R Core Team, 2020). We employed a linear mixed effects model using the 'nlme' package in the R software (Pinheiro et al., 2020) to test the impact of deep water component (inorganic, organic, biology) and nutrient level (high/low nitrate). The linear model is robust against missing data points, meaning a consistent test could be employed across all dependent variables and heteroscedasticity (variability within replicates) could be taken into consideration as a result through a variable model variance structure in the linear mixed model. The initial model (fixed effects = deep water component * nutrient level * sampling day, random effects = ~1 | Incubation bottle) was simplified stepwise to retain only the terms that remained significant to the model result. Nutrient level, deep water component, and sampling day were all included as factors (non–continuous) as independent variables against the continuous dependent variable. The model simplification was applied to the bloom period (Day 1–4, until peak Chl $a$ concentrations) and post-bloom period (Day 5–10) separately due to the non–monotonic response and distinct biological responses between the two periods. The contrast matrix used is reported in Table 2 and shows that "organic" was used as the control for the linear mixed model analysis. The contrast matrix hence means that reported model significance hence refers to the difference between organic vs. biology and organic vs. inorganic to distinguish the biological treatment effect and the organic effect.

**Table 1. Contrast matrix used to compare treatment effects in the linear mixed model applied to the bloom and post-bloom periods.**

| Deep water component | Contrast 1 | Contrast 2 |
|---|---|---|
| Biology (deep water) | 1 | 0 |
| Organic (filtered deep water/control) | -1 | -1 |
| Inorganic (surface water) | 0 | 1 |
| Sum of contrasts | 0 | 0 |

The impact of random effects was tested for both incubator and bottle number with no significant impact of either on the model. Where the QQ plot indicated that extreme values skewed the model and to resolve heteroscedasticity, log(fixed effects) was employed and the linear model re-simplified as described above. These results are reported for example as log(Chlorophyll), (see Supplementary Material). Outliers in model fit were identified in initial model fit and excluded from the final model fit when the residuals were outside the 95% confidence interval (CI). Identified outliers were solely detected and excluded in nutrient drawdown ratios ($\Delta NO_x{:}\Delta DIP$). Post hoc tests were carried out on the interaction term using False Discovery Rate ("fdr") with q = 0.05 in the package 'emmeans' in R software (Lenth, 2020). In general, time was considered relevant for the emergence of effects as the bloom developed but was not considered an experimental factor. Hence post hoc test output is reported primarily for nutrient level (high nitrate or low nitrate) and deep water component (inorganic nutrients, biology or organic as the control).

A non-parametric analysis of similarity test (ANOSIM) was carried out to determine if the difference in phytoplankton composition between the treatments (among group similarity) was smaller or larger than that between treatment replicates (within group similarity). Data were grouped by bloom status (pre-bloom = Day 1–2, bloom = Day 3–4, post-bloom = Day 8–10) and differently to the linear mixed effect model applied, to enable detection of treatment-related differences in initial phytoplankton composition. A Bray-Curtis dissimilarity matrix was constructed and the stress was calculated and accepted if less than 0.2, using the 'vegan' package in the R software (Oksanen et al., 2019). SIMPER (similarity percentages) were calculated post hoc on the Bray–Curtis distance matrix to distinguish influential groups behind the detected numerical dissimilarity.

## 3 Results

### 3.1 Initial conditions and initial treatment differences (Day 1)

Overall, initial nitrate ($NO_x$) and phosphate (DIP) concentrations indicated successful implementation of the experiment design for the two nutrient levels. Initial nitrate ($NO_x$) concentrations were similar in the high nitrate (HN, $[NO_x]$ = 7.72 ± 0.46 µmol $L^{-1}$, mean ± s.d., n = 12) and in the low nitrate treatment levels (LN, $[NO_x]$ = 2.56 ± 0.54 µmol $L^{-1}$ mean ± s.d., n = 12, Fig. 3B). Importantly, within each nutrient level, there were minor differences in $NO_x$ between the three deep water components (Table S1). Inorganic phosphate concentrations were ~2 µmol $L^{-1}$ in all six treatment combinations (Fig. 3C).

A proxy used to indicate dissolved organic carbon concentrations ($a_{254}$) showed small differences between the treatments only with surface water ("inorganic"), and those containing filtered or unfiltered deep water ("organic", "biology", Fig. 4C, Table S1). Highest initial values were observed in the "inorganic" incubations and lowest concentrations in low nitrate treatments for both "organic" and "biology", which both contain deep water. This was likely due to lower DOC concentrations in the deep water collected than the mesocosm water. No clear initial differences between nutrient levels or deep water components were observed in the proxy of dissolved organic matter (DOM) molecular weight ($E_2$:$E_3$, Fig. 4D, Table S1). Leucine aminopeptidase (LAP) activity, which indicates organic nitrogen remineralisation, was slightly higher in the surface water ("inorganic") incubations than in the "organic" or "biology" incubations. This could be a residual signal due to inorganic nitrogen depletion in the surface mesocosm water that was not diluted by the deep water nutrients added (Fig. 3F, Table S1). An overview of variables for the two deep water treatments on Day 1 is also provided in Table S1 (Supplementary Material).

Mean Chl $a$ concentration was initially similar among all treatments and ranged between 2.61 ± 0.79 and 3.41 ± 0.76 µg $L^{-1}$ (mean ± s.d., n = 4, Fig. 3A), indicating similar starting phytoplankton biomass. Cell abundances of two key phytoplankton groups identified by flow cytometry, *Synechococcus* and nanoplankton, were similar across all six treatment combinations (Fig. 5). Initial phytoplankton community composition based on relative contribution of each group to Chl $a$ fluorescence in flow cytometry analyses (Table S1) did indicate variability between replicates within a treatment, but this was not related to either deep water nutrient status (high/low nitrate; R statistic = 0.0313, p = 0.131) or the treatment ("inorganic"/"organic"/"biology", R statistic = 0.0212, p = 0.276). Abundances of large microphytoplankton were much lower than for other groups such as small microphytoplankton, and ranged between 0 and 11 counts per sample (analysed volume = 100 µL). Nevertheless, we included these in the community composition due to their large size and contribution to the Chl $a$ fluorescence signal, but focused our attention on more dominant groups where we consider the underlying data to be more robust. The maximum quantum efficiency of photosystem II ($F_v/F_m$) also presented small differences at the beginning of the experiment (0.49 ± 0.01, mean ± s.d., n = 24, Fig. 4B, Table S1), that alongside with no differences observed in Chl $a$ and community structure, suggest that phytoplankton initial conditions were similar in all treatments.

**Figure 3: Measured A) Chlorophyll $a$ concentrations, B) nitrate + nitrite concentrations, C) phosphate concentrations, D) silicate drawdown relative to initial concentrations on Day 1 (Eqn. 2), (E) calculated nutrient drawdown stoichiometry (nitrate+nitrite consumed vs. phosphate consumed, Eqn. 1), and (F) measured leucine aminopeptidase (LAP) activity over the 10 day study period. Dots indicate means across four treatment replicates and the error bars indicate the corresponding calculated 95% confidence interval for each sampling day. The dashed line in (E) indicates the Redfield ratio of 16:1.**

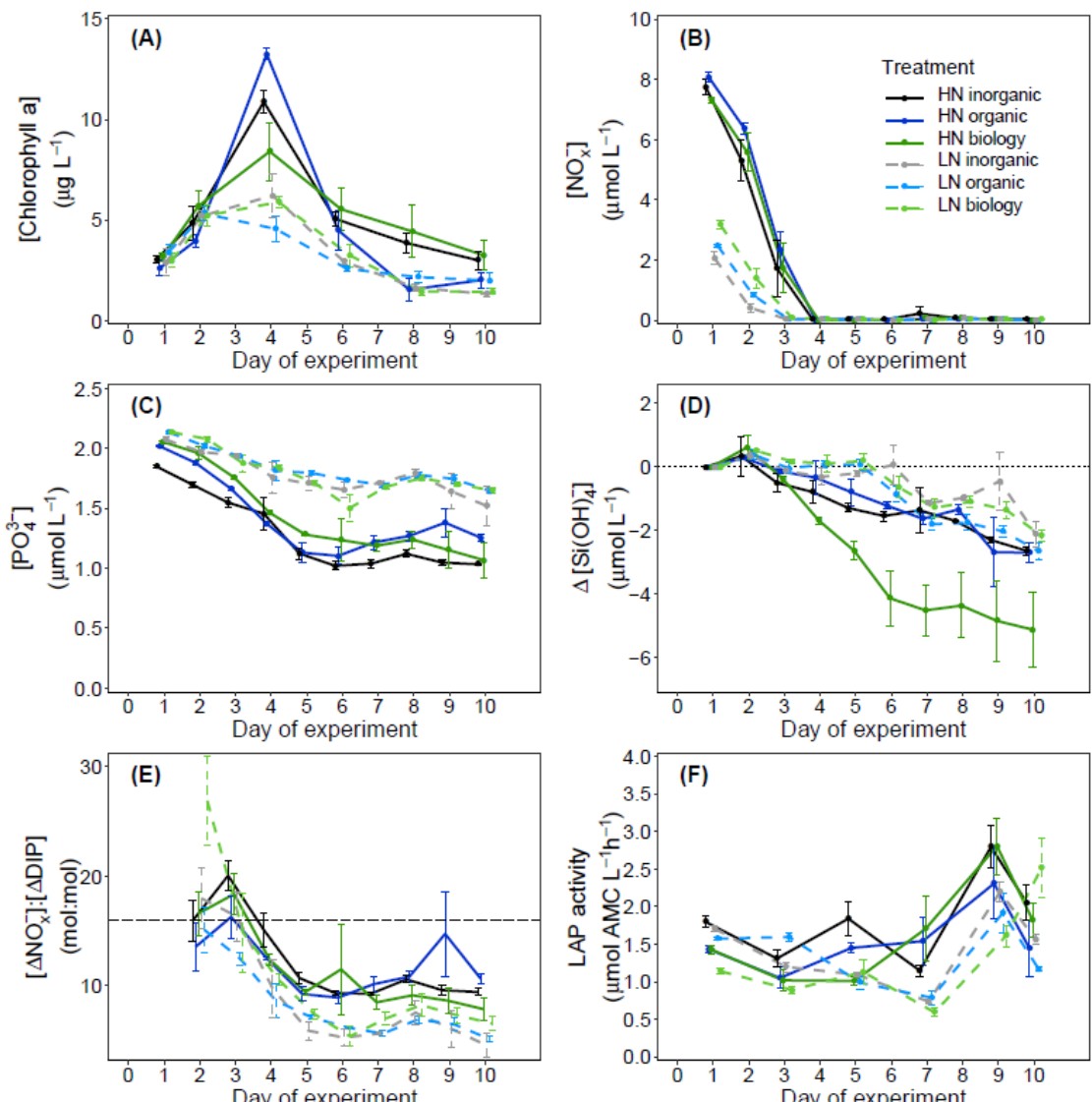

## 3.2 Bloom phase (Day 2 – Day 5)

Inorganic nitrogen ($NO_x$) was rapidly consumed in all incubations with concentrations reaching below analytical detection limits within 3–4 days (Fig. 3B). DIP concentrations also decreased between Days 2 and 5 (Fig. 3C). Non-Redfield nutrient utilisation was observed after $NO_x$ depletion (Fig. 3E). Significant differences in $\Delta NO_x$:$\Delta DIP$ emerged over time in the bloom phase between the nutrient levels ($p = 0.0025$, F-value = 6.9759, Table S3a) and between deep water components ($p = 0.0456$, F-value = 3.6185, Table S3a). Although initial dissolved silicate concentrations were different among the treatments, concentrations were not limiting and remained above 2.5 µmol L$^{-1}$. To more easily detect any differences between the treatments, we then calculated the drawdown in dissolved silicate concentrations ($\Delta DSi$, Eqn. 2, Fig. 3D). This indicated that silicate drawdown was highest in the high nitrate "biology" treatment between Days 3 and 5.

Chl *a* concentrations increased during this bloom phase with peak concentrations of up to 12 µg L$^{-1}$ and ~6 µg L$^{-1}$ for high nitrate and low nitrate, respectively, observed on Day 4 (Fig. 3A). $F_v/F_m$ responded similarly to Chl *a* concentration with higher values (lower cellular stress) in high nitrate than in low nitrate incubations. Flow cytometric analysis showed that nanoplankton cell abundances also peaked around Day 4 in most incubations (Fig. 5D-F). Treatment differences were also were observed in Chl *a* concentration in the bloom period (p = 0.0026, F-value = 5.0041, Table S2a). Post hoc tests indicated that these differences were due to a significant effect of nutrient level ("HN" – "LN") on Chl *a* concentration in both the "organic" and "inorganic" incubations as well as a significant effect of the deep water biology (comparison between "biology" – "organic") on Chl *a* in the LN incubations (Table S2b).

While nutrient level did not have a significant effect on bloom phytoplankton community composition, the three deep water components did (ANOSIM, R = 0.2214, p = 0.001, Table 2) with the highest dissimilarity detected between the "biology" and "organic" treatments, primarily due to differences in the nanoplankton group (Table 2, Fig. 5D-F). LAP activity during the bloom was similar to initial rates (1.0 – 2.0 µmol AMC L$^{-1}$ h$^{-1}$, Fig. 3F) and where differences in activity between treatment levels were detected, these were higher in the high nitrate incubations (Day 5, Table S4b). Average phytoplankton cell size, indicated by mean forward scatter (FSC-A) from flow cytometric analyses, increased to reach a maximum on Day 5 (Fig. 4A) coinciding with inorganic nitrogen depletion.

**Figure 4: A) mean forward scatter (FSC-A) from flow cytometric analyses as a proxy of relative phytoplankton cell size during the study period, B) maximum photochemical efficiency ($F_v/F_m$), C) absorption coefficient at 254 nm ($a_{254}$) used as a proxy for DOC concentration, and D) the ratio of the absorption coefficients at 250 nm and 365 nm ($E_2:E_3$) as an indicator of DOM molecular weight.**

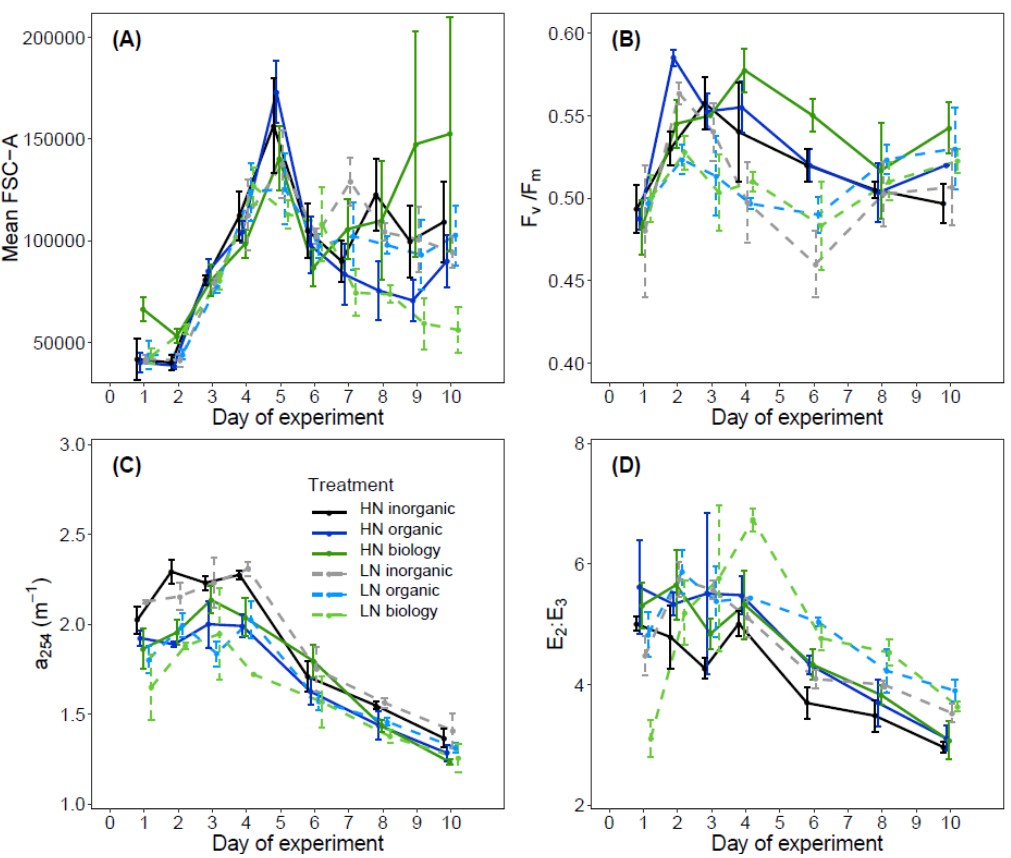

## 3.3 Post-bloom phase (Day 6 – Day 10)

In this phase, inorganic nitrogen ($NO_x$) concentrations remained low or below detection limit. DIP concentrations remained relatively constant or even increased slightly between Day 6 and Day 8 (Fig. 3C). Treatment related differences in nutrient uptake ratio were also detected in the post-bloom phase with over 50% higher consumption of DIP compared to $NO_x$ in low nitrate than in high nitrate incubations by the end of the 10-day long study period ($\Delta NO_x$:$\Delta DIP$ on Day 10: high nitrate = 9.32 ± 1.68, low nitrate = 5.43 ± 1.61, mean ± s.d., n = 6, see also Fig. 3E). Despite very low nitrate concentrations, dissolved silicate continued to be consumed (Fig. 3D). This indicated sustained growth of silicifying phytoplankton species during the post-bloom phase, even though overall nanoplankton abundances had decreased after their bloom phase peak in most incubations (Fig. 5D-F). Chl *a* concentrations decreased compared to the bloom phase, reaching similar concentrations on Day 10 as the initial measurements on Day 1, hence indicating decline in phytoplankton biomass (Fig. 3A) and a post-bloom status of the plankton community.

**Figure 5: Abundances of two key phytoplankton groups identified in flow cytometric analyses, *Synechococcus* (A-C) and nanoplankton (D-F), measured daily over the 10-day long study period. A divergent response was observed between "biology" incubations in both phytoplankton groups during the nutrient-depleted post-bloom period after Day 4.**

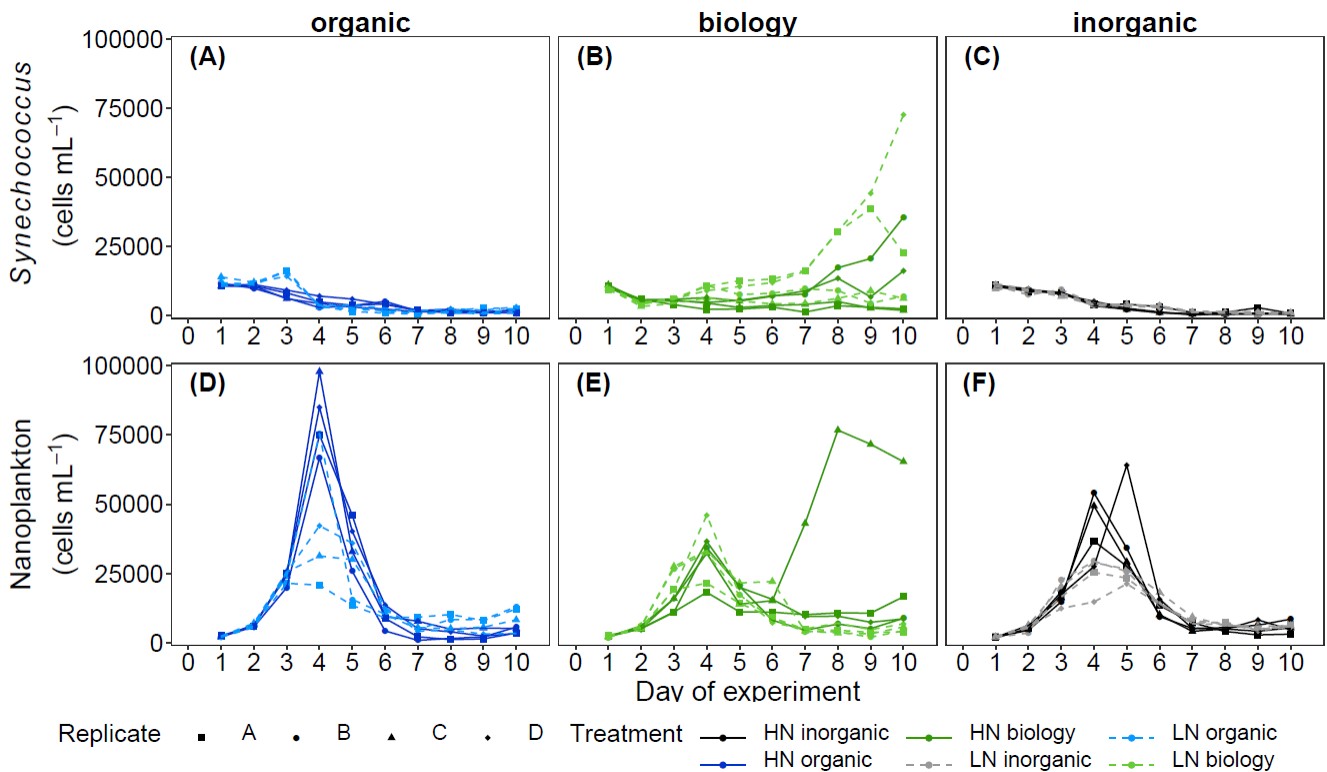

Chl *a* concentrations were significantly different between the nutrient levels in the "inorganic" and "biology" incubations on Day 8 and Day 10 (Post hoc Tukey Pairwise comparison, "biology": $p_{adj}$ = 0.0012, $p_{adj}$ = 0.0129; "inorganic": $p_{adj}$ = 0.051, $p_{adj}$ = 0.0087, for Day 8 and Day 10, respectively. See also Table S2b in Supplementary Material). Phytoplankton community composition was also influenced by both nutrient level and deep water component in the post-bloom phase, although both effects were weak (R = 0.04404, R = 0.07601, respectively) with an overall dissimilarity between the three deep water components of 35–37% (Table 2).

**Table 2: Results of one-way ANOSIM analyses and Post hoc SIMPER analyses reporting only significant differences in measured cell abundances, where detected, and the groups with the highest and significant contribution to these treatment differences. Bloom phase is defined here as Day 3 and Day 4, and Post-bloom is defined as Day 8 to Day 10.**

| | Factor | R statistic | p value | Comparison | Overall dissimilarity between treatments | Key group | Contribution to detected treatment differences |
|---|---|---|---|---|---|---|---|
| **BLOOM** | Deep water component (inorganic, organic, biology) | 0.2214 | 0.001 | Biology vs. Inorganic | 16.70% | Chains | 3.37% |
| | | | | Biology vs. Organic | 19.02% | Nanophytoplankton | 7.57% |
| | | | | | | Chains | 3.37% |
| | | | | | | Picoeukaryotes | 1.75% |
| | | | | Inorganic vs. Organic | 16.05% | Large microphytoplankton | 3.61% |
| | | | | | | *Synechococcus* | 0.46% |
| **POST-BLOOM** | Nutrient level | 0.04404 | 0.037 | High nitrate vs. low nitrate | 36.29% | Nanophytoplankton | 10.03% |
| | | | | | | *Synechococcus* | 0.06% |
| | Deep water component (inorganic, organic, biology) | 0.07601 | 0.005 | Biology vs. Inorganic | 36.92% | Small microphytoplankton | 11.12% |
| | | | | | | Synechococcus | 0.99% |
| | | | | Biology vs. Organic | 37.17% | Picoeukaryotes | 4.28% |
| | | | | | | Chains | 3.93% |
| | | | | | | *Synechococcus* | 0.94% |
| | | | | Inorganic vs. Organic | 35.28% | FL4A | 0.17% |

Divergence in average phytoplankton cell size between treatments occurred during the post-bloom period (Fig. 4A) and some treatment differences in the abundance of key phytoplankton groups (*Synechococcus*) also emerged. *Synechoccocus* abundances increased in both the low and high nitrate "biology" treatments during the post-bloom period (Fig. 5A-C). This increase occurred in low nitrate treatment where initial nitrate concentrations were lower and presumably phytoplankton reached nitrate-limited growth earlier. There was an increase in average cell size in the high nitrate "biology" incubations (Fig. 4A), in addition to particularly high post-bloom silicate consumption in the high nitrate "biology" incubations as well marked differences in silicate uptake between the replicates in the high nitrate "biology" incubations (see large error bars in Fig. 3D). This high variability was driven by the response of one high nitrate "biology" replicate that also had the highest phosphate drawdown, and post-bloom Chl a concentration and also had highest nanophytoplankton abundances (Fig. 5E).

# 4 Discussion

## 4.1 Peak bloom biomass was affected by seed population in upwelled deep water

As expected, nutrient addition from the deep water stimulated surface phytoplankton biomass production and increased average
phytoplankton community cell size. This fits well with the general understanding of phytoplankton blooms in upwelling regions where larger phytoplankton, often diatoms, dominate bloom biomass as sporadic wind driven upwelling events bring nutrient rich subsurface water to the photic layer (Sydeman et al., 2014). In this study, higher initial nitrate concentrations generally lead to higher peak phytoplankton biomass and higher photosynthetic energy conversion efficiency ($F_v/F_m$), as expected (Falkowski et al., 2017). Hence, we consider the availability of inorganic nitrogen to be a primary factor controlling
organic matter production in this truncated food web (<64 µm).

However, while nitrate concentrations were the same between high nitrate levels, bloom development was not. There were noticeable differences in peak bloom Chl $a$ concentrations between the high nitrate level incubations ("organic", "inorganic" and "biology") of up to 6 µg $L^{-1}$ Chl $a$. In particular, the unfiltered deep water incubations (high nitrate "biology"), testing the impact of the seed microbial community, had lowest peak Chl $a$ concentrations. Sharper bloom biomass development in the
filtered high nitrate "organic" and high nitrate "inorganic" treatments suggest a primarily bottom-up driven food web response to nutrient addition. Bloom development in high nitrate "biology" was more muted as nutrient competition within the plankton community (e.g. with heterotrophic bacteria) was likely higher, due to the lack of organism dilution compared to the high nitrate filtered "organic"/"inorganic" nutrient treatments. Alternatively, this muted biomass development could suggest an increase in grazing pressure via potential introduction of microzooplankton grazers (<64µm) in the addition of unfiltered deep
water. Hence, higher retention of Chl $a$ post-bloom in this treatment suggests potentially longer sustained periods of productive biomass when deep water plankton are added concurrently with upwelled nutrients. The precise mechanism(s) underlying this response, however, requires further detailed elucidation.

Among the four replicate "biology" incubations with the same measured initial nutrient concentrations and proportions of surface and deep water and incubation light and temperature, there were also marked differences in maximum Chl $a$
concentrations that could not be explained by the amount of added nitrate. Divergence in Chl $a$ biomass within a given treatment (i.e. with the same initial nitrate concentrations) occurred after nitrate concentrations were below detection around Day 4. This indicates that flexible nitrogen assimilation strategies were employed by the same starting community to produce active primary producing biomass: nitrate was either internally stored in phytoplankton cells (Bode et al., 1997) and could not be detected in filtered nutrient analyses or alternative nitrogen sources were utilised e.g. dissolved organic nitrogen or rapid
ammonia assimilation thereby supporting higher Chl $a$ biomass. Yet more importantly, these strategies must have been employed to a different degree of success. This variable response to nutrient additions contributes another layer of complexity when projecting primary producer responses to upwelling in the Peruvian Humboldt Current System. The variability between both the filtered ("organic") and unfiltered ("biology") deep water incubations, in addition to the variability between replicates in the unfiltered deep water incubations, suggest that the microbes present in subsurface waters are key drivers in the observed
biomass response to upwelled waters in the euphotic zone.

Despite high variability in the biomass response – both between and within the six treatments – the consumption of excess phosphate (i.e. degree of non-Redfield nutrient uptake) was more dependent on the initial nitrate concentrations rather than the deep water biology. Our results also indicate that over 50% more phosphate was consumed per mol of $NO_x$ in the low nitrate treatment (mean initial $NO_x$:DIP = 1.21 ± 0.24), than with high nitrate (mean initial $NO_x$:DIP = 3.92 ± 0.32). Phosphate was
never depleted in this study. Biological dinitrogen fixation was likely a minor source of new nitrogen compared to nitrate inputs during this time, as was found in the parallel mesocosm study (Kittu et al., n.d.). We also have no evidence from enzyme rates measured, that there was any stimulation of microbial nutrient regeneration to satisfy N demand, despite N–depletion. We had anticipated an upregulation in leucine aminopeptidase (LAP) activity, a protein-hydrolysing enzyme, where inorganic nitrate concentrations were lowest and hence most limiting in the low nitrate treatments. Instead, LAP activity was highest
during the post-bloom period and in the high nitrate incubations where there was more semi-labile organic matter present, as indicated by higher signatures in $a_{254}$ (a proxy of DOM, Fig. 3F) and $E_2$:$E_3$ (a proxy of more labile high molecular weight compounds, Fig. 4C,D) (Benner and Amon, 2015). The LAP activity was one to two orders of magnitude higher than most literature values. For example, in a study from the same region but further from shore, the LAP activity was 20-65 nmol AMC

L$^{-1}$ h$^{-1}$ in natural communities (Maßmig et al., 2020). Partly, the high LAP activity in this study could be due to the high concentration of substrate we used (500 µmol L$^{-1}$ leu-AMC), which aimed to measure maximum hydrolysis rates. However, this cannot be the only reason for the high values. For comparison, we used only 2.5 times higher substrate concentration compared with Maßmig et al. (2020). The high LAP activity and close relationship with fresh, labile organic matter production suggests that LAP was produced to support bacterial production above the oxycline (Loginova et al., 2019) rather than compensating for higher N–limitation in the low nitrate incubations.

**4.2 Seeding of deep water populations is a key driver for plankton succession and biogeochemistry in surface blooms**

We expected to have similar initial plankton assemblage composition in the "organic" and "inorganic" nutrient incubations as all plankton present originate from the surface mesocosm water. Differences in initial composition would be expected in the "biology" incubations due to addition of deep water communities from two different locations and depths, but these were not significant in this study and no clear difference in initial abundances between treatments was detected in phytoplankton according to flow cytometry data. Phytoplankton assemblage composition across all six treatments converged during the bloom as rapid growing groups, probably silicifying phytoplankton such as diatoms based on flow cytometric size class and observed silicate consumption, dominated overall biomass. Inherent variability in plankton community dynamics between treatments and among replicates was revealed in nanoplankton and in *Synechococcus*, when resource availability (here, nitrate) limited net growth after Day 4. Post-bloom community composition on Day 10 was affected by nutrient level and deep water component although treatment differences were small and variability between replicates within a treatment emerged. This variability may have originated in the initially enclosed microbial populations, even though we designed the study to minimise heterogeneity by pooling all treatment water and continuously mixing the tanks while randomly filling the replicate incubations. Hence, not only the absolute biomass concentrations as previously discussed, but also phytoplankton community composition post-bloom was determined by the seed microbial populations initially present.

We propose that different mechanisms drove the divergent response of phytoplankton community composition between treatments and between replicates, in particular for two key groups: *Synechococcus,* and nanoplankton/chain-forming species that were likely diatoms based on the magnitude of dissolved silicic acid consumption. Diatoms were likely beneficiaries of nutrient addition as they are considered "transcriptionally proactive" (Lampe et al., 2018). This means they can respond quickly and take advantage of nitrogen resources when sporadically available, for example during upwelling (Fawcett and Ward, 2011; Stolte and Riegman, 1995) or after nutrient inputs into nutrient poor surface waters in oligotrophic gyres (Lampe et al., 2019). Rapid nutrient uptake and growth by diatoms lead to their ecosystem dominance in nutrient–initiated phytoplankton blooms such as those in coastal upwelling systems (Lassiter et al., 2006). Silicate consumption post-bloom and more nanoplankton species in unfiltered "biology" incubations suggest resting spores or down welled chain-forming silicifying phytoplankton were indeed present in the subsurface waters, thriving when irradiance levels increased upon incubation. Moreover, distinct responses of the silicifying phytoplankton between the two deep water sources in the unfiltered "biology" treatments is further evidence that the seed population in upwelled waters modulates surface bloom dynamics of diatom populations in the Humboldt Upwelling System.

Initial cell abundances were similar in all treatments, and ANOSIM analyses did not detect any significant differences in community composition, thus the majority of the starting community stemmed from the mesocosm surface water, rather than the manipulated deep water. Lack of net *Synechococcus* growth in the "organic" treatments but growth in the unfiltered "biology" treatments with the same seawater chemistry (i.e. inorganic and organic nutrients and trace metals) points towards a mutually beneficial relationship, either metabolic or ecological, between *Synechococcus* and an unidentified member of the unfiltered deep water plankton community (< 64 µm). For example, a change in dominant predators upon addition of deep water may have relieved grazing pressure on these picocyanobacteria. Alternatively, a metabolic response could be due to underlying induced changes in gene expression (Robidart et al., 2019) or a dependency (syntrophy) with a deep water microbe/organism may have evolved, selectively supporting the cooccurrence of *Synechococcus* and other microbes through complementary metabolic function (Morris et al., 2012). There is also evidence that viral presence and lysis of heterotrophic bacteria may also enhance *Synechococcus* growth (Weinbauer et al., 2011). Slow growing picocyanobacteria lend themselves more to stable mutualistic relationships than faster growing diatoms that quickly consume resources and generally follow a 'boom or bust'-like biomass trajectory. Hence, the different physiological response times – rapid in diatoms and comparatively

slower in picocyanobacteria – appears to underlie the variability in biomass observed in this incubation experiment. The bloom community contained higher diatom abundances, driven by initial but immeasurable differences in seed community enclosed, and the sustained differences emerging post-bloom in *Synechococcus*. The slower but more consistent growth of *Synechococcus* may indicate why these picocyanobacteria are often observed post-bloom and in more oligotrophic waters further offshore (Franz et al., 2012), even though the origins of their biomass can be apparently influenced by upwelling of nutrient rich water and the microbial communities contained within close to the coast.

The lack of consistent response in multiple variables (Chl *a*, silicate concentrations, phytoplankton abundances) across all "biology" incubation replicates further shows how heterogeneity in subsurface water seed communities can shape the resulting plankton bloom development, biomass accumulation and nutrient concentrations in surface waters. In addition to initial nitrate concentrations, the seed microbial population variability impacted final nutrient stoichiometries and in particular silicate:nitrate utilisation. Over 2.4 µmol L$^{-1}$ more silicate was consumed after nitrate was exhausted and higher post-bloom Chl *a* concentrations and nanoplankton abundances were sustained in the one HN "biology" replicate compared to the three other replicates.

### 4.3 Consequences for phytoplankton succession and productivity after upwelling events

Natural variability in composition or fitness in initial plankton assemblages can bring about significant variability in measured response variables that can be larger than those driven by the experimental treatment itself (Krishna and Schartau (2017)). Yet it is surprising for such a strong driver of phytoplankton succession, like nitrate concentrations, that the deep water biology had such a measurable influence. A similar study in the southern (Chilean) Humboldt Current System investigated the impact of N:P ratios on different surface communities and came to the same conclusion: initial community composition was more important than inorganic N:P ratio for food web functioning and biogeochemistry (Spilling et al., 2019). Stable and consistent relationships between nutrient availability, consumption and the produced biomass are clearly not a feature of this dynamic ecosystem. The vital role of the seed population in modifying the bloom following upwelling events may even be a general characteristic of plankton communities in the Humboldt Current System, and other upwelling ecosystems.

Hence, nutrient upwelling promotes bloom development and phytoplankton growth while the upwelled community modulates these blooms in composition. This makes prediction of coastal phytoplankton productivity a particular challenge as the entrainment of subsurface populations depends on depth and rates of water mass transport and modulation of source water in either depth or location will likely reflect in altered ecology of bloom populations in coastal waters, and potential biogeochemical changes in nutrient cycling. This would be in addition to the variability in nutrient content of upwelled water sources via ENSO (Espinoza–Morriberón et al., 2017) and variability in the mixed layer depth (Rigby et al., 2020). It is even possible that this uncertainty would be amplified if this response occurred in a more complete food web with larger predators. Smaller organisms e.g. *Synechococcus*, that are less important for determining high biomass during the blooms, did not have clear impact on nutrient stoichiometry within the study time period despite clearly profiting from the addition of a deep water microbial community. Indeed, vertical mixing through Ekman pumping away from the coast may even provide sporadic stimulation of surface *Synechococcus* populations as water masses are advected offshore into the South Pacific subtropical gyre. In the ocean, physiological and ecological drivers (e.g. growth rates, transcriptional response times, mutualisms, symbioses, Sect. 4.2) would act in addition to other physical factors that regulate plankton biomass accumulation and succession in the surface waters following upwelling (e.g. seed community abundance present in subsurface waters; Seegers et al. (2015)). Such physical factors, such as dilution, mixing and horizontal transport of water masses via regular tidal transport onshore (Stauffer et al., 2020), could not be included in this experimental set-up.

## 5 Conclusions

Overall, productivity, i.e. organic matter production in this 10-day long incubation study, was highest in incubations with the highest added nitrate concentrations, reflecting N as the limiting macronutrient. Differences in Chl *a* concentrations were primarily connected to the amount of nitrate added, but were also distinctly modified by the seed community added. Incubations with different initial source water showed that the microbial seed community impacts phytoplankton succession,

with the potential to influence communities further offshore and towards the oligotrophic subtropical gyre. The crucial period for differences in microbial community was the nutrient depleted post-bloom phase, where increased resource competition elicited divergence in biomass and post-bloom composition between replicates in unfiltered incubations. These differences in community composition had an impact on nutrient drawdown. For example, silicate drawdown was higher in the unfiltered "biology" incubations compared to filtered "organic" incubations (within a given deep water). This indicates potential differences in diatom growth between replicates, likely arising from a seed population of diatoms in the unfiltered deep water added. Initial minor heterogeneity in microbial community composition, such as that observed in diatoms and *Synechococcus* here, may be augmented in further successions of plankton bloom developments and have consequences on overall productivity and transfer of energy to higher trophic levels. Hence, nutrient upwelling promotes the occurrence of phytoplankton blooms while the upwelled community modifies these blooms.

## Data availability

All data presented in this manuscript are openly available on PANGAEA under the following link: https://doi.org/10.1594/PANGAEA.941138

## Supplement link

Detailed statistical output and other data referred to here in the text is available as Supplementary Material.
Table S1: Overview of key variables on Day 1

Table S2a: Fixed effects for final simplified model for log-transformed Chl *a* concentrations
Table S2b: Post hoc comparisons for Chl *a* concentrations

Table S3a: Fixed effects for final simplified model for $NO_x$:DIP drawdown
Table S3b: Post hoc comparisons for $NO_x$:DIP drawdown

Table S4a: Fixed effects for final simplified model for leucine aminopeptidase (LAP) activity
Table S4b: Post hoc comparisons for leucine aminopeptidase (LAP) activity.

Figure S1: Relationship between PAR and measured Lux
Figure S2: Flow cytometry cytograms to indicate gating of different groups.

## Author contributions

This study was conceived and designed by AJP, JA, KS, LTB, UR. Sampling and sample analysis were carried out by AJP, EvdE, JA, JP., KS, LR, LTB, and NH. Data analysis was carried out by AJP, JA, JP, KS, LR, LTB, NH with the manuscript written by AJP with comments from all coauthors.

## Competing interest

The authors declare that they have no conflict of interest.

**Special issue statement**

This manuscript is submitted to be published as a part of a Biogeosciences Special Issue "Ecological and biogeochemical functioning of the coastal upwelling system off Peru: an in situ mesocosm study".

**Acknowledgments**

This project was supported by the Collaborative Research Centre SFB 754 Climate–Biogeochemistry Interactions in the Tropical Ocean financed by the German Research Foundation (DFG). Additional funding was provided by the EU project AQUACOSM and the Leibniz Award 2012 granted to U.R. L.R. acknowledges the support of ANID-CENTROS REGIONALES R20F0008 and FONDECYT Project #3170156. L.T.B acknowledges funding from the Australian Research Council (FT200100846), and KS and JP funding from the Academy of Finland (259164). J.A. was supported by a Helmholtz International Fellow Award, 2015 (Helmholtz Association, Germany). We are particularly thankful to the staff of the Instituto del Mar del Perú (IMARPE) for their support during the planning, preparation and execution of this study and to the captains and crews of BAP MORALES, IMARPE VI and BIC HUMBOLDT for support during deployment and recovery of the mesocosms and various operations during the course of this investigation. Special thanks go to the Marina de Guerra del Perú, in particular the submarine section of the Navy of Callao, and to the Dirección General de Capitanías y Guardacostas. This work is a contribution in the framework of the Cooperation agreement between the IMARPE and GEOMAR through the German Ministry for Education and Research (BMBF) project ASLAEL 12–016 and the national project Integrated Study of the Upwelling System off Peru developed by the Direction of Oceanography and Climate Change of IMARPE, PPR 137 CONCYTEC. We also thank Tim Boxhammer for advice on figure preparation and design.

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
