# Peer review of "Upwelled plankton community modulates surface bloom succession and nutrient availability in a natural plankton assemblage"

_Biogeosciences, 2022_

## Author Response (AR1)

**Author response to comments provided by Referee #1 (Citation: https://doi.org/10.5194/bg-2022-44-RC1) including a list of all relevant changes made in the manuscript corresponding to each referee comments.**

We thank the Referee for their constructive comments and suggestions and respond to these point-by-point below. Our comments are presented in *italics*.

Additional note: We added the details of any changes made to the manuscript below our initial comments in **bold** ("Corresponding changes made"). All line numbers in our responses refer to the revised manuscript (no track-changes).

**General comments**

The manuscript shows a nice experiment planned to demonstrate the influence of the chemistry and biology of upwelled water on the development and later progress of phytoplankton blooms in coastal upwelling systems. In my opinion, the design of the experiment is correct, in which the simulation of the dilution caused by an upwelling episode stands out. However, upwelling most likely does not cause a full 1:1 mixing of upwelling water with surface water. Usually, upwelling pushes up, compresses the surface layer, and so supplies nutrients (and plankton) by diffusion and turbulent mixing at different intensities.

The conclusions are correct, although expected. The first conclusion is the best known. That is, nutrient supply to surface water with low nutrient concentrations induces phytoplankton blooms, mainly diatoms. The second is somewhat new, but not strange. It is reasonable to expect that plankton populations reaching the surface with upwelled waters modulate the bloom and its later evolution. This experiment clearly demonstrates that this happens. However, a better characterization of the species and/or genera of phytoplankton involved is lacking. The flow cytometer has only allowed the characterization of Synechococcus. For the rest of the community there was only a proxy of its size with very low detection of microphytoplankton. In addition, chlorophyll was not fractionated. This information is especially important in the post-bloom, when divergence between treatments and the variability within treatments is more evident. However, there is also variability among treatments during the bloom, as inferred from the differences in chlorophyll concentration on day 4 (Fig. 3A) and in the different abundances of nanoplankton (Fig. 5D, E, F). On the other hand, the results reporting significant silicate drawdown in the HN biology treatment point to the importance of diatoms, which could be different from those found in other treatments, including the LN biology treatment. Microphytoplankton (mainly diatoms) are likely causing the divergence observed in both bloom and postbloom.

Although the introduction and the discussion read well, this is not the case for the results. In my opinion, this section, of great importance to support the conclusions, is written in a cumbersome way. It is necessary to read it several times and with enough attention to catch the information. Figures are not always properly cited, nor is supplementary material. There are tables in the supplementary material that are not cited in the text. In my opinion, this results section could be improved to remove weaknesses and make the manuscript more attractive to potential readers. The manuscript will probably improve by focusing the description of results on those relevant to the conclusions and ignoring those with low contribution to the two main conclusions.

Despite the lack of information on phytoplankton species composition, which in my opinion represents the greatest weakness of the manuscript, the design of the experiment and the difficulty of its execution, lead me to recommend the publication of the manuscript with major revisions.

Author response: We thank the reviewer for their constructive comments and acknowledge the feedback on the cumbersome results section. We would revise the methods section to improve clarity, focus more on the key results presented in the discussion, ensure accurate citation of figures and that all supplementary materials are appropriately cited in the manuscript. A number of comments from Reviewer #2 will also help in revising these sections (please also see our responses to these comments).

We also agree with the reviewer that a better characterisation of the phytoplankton community would be preferable, this is also a point raised by Reviewer #2, and we acknowledge this as a limitation of our study. We made this decision based on practical reasons and limitations in the experiment set-up, in particular, the sampling volumes necessary and the space available for incubations. While the information in flow cytometric analysis does not enable characterisation of species, it does provide functional information such as size, fluorescence, etc that can be useful in interpreting observations with the benefit of requiring much smaller sample volumes. This was a particular advantage for our experimental set-up that did mean we sacrificed some more detailed information on the plankton community. Fractionating chlorophyll would be another way to determine the size structure of the phytoplankton community but again requires quite large volumes compared to flow cytometry to achieve this.

Not only Synechococcus was characterised in this study, but indeed certain key groups by size/fluorescence (not at the species/genera level). Groups, where a treatment effect was clearly observed, were highlighted to focus the manuscript and the key conclusions. Future experiments would certainly benefit from this knowledge of significant responses in this study to understand which analyses should be incorporated to better understand the biological drivers of phytoplankton bloom initiation and succession.

Corresponding changes made: We have made significant changes to both the methods and results section to ensure that the treatment distinctions and various methodological steps are more clearly distinguishable based on the names. The figure indicating the experimental design was remade to address the referee comments. We also took care in the results section to only report essential data to support the main conclusions and the discussion and the ensure that the relevant figures and tables were cited appropriately and removed text where no longer necessary.

**Specific comments**

**Introduction**

Line 29. ...is considered the most productive... from where? Maybe ..."the most productive upwelling region" or something similar

Author response: We would use the suggested "... the most productive upwelling region in terms of fish yield..." for a revised manuscript.

Corresponding changes made: Please see line 28 in the revised manuscript. This was modified to read "...most productive upwelling region in terms of fish production ...".

Lines 66-68. The Peruvian productivity paradox is a common paradox to all upwelling systems. With strong upwelling, chlorophyll concentration is low because surface water is recently upwelled water with high nutrient concentrations. Chlorophyll concentration increases when upwelling relaxes. This time lag also has translation into spatial heterogeneity. Chlorophyll concentration is low (few phytoplankton) in the upwelling center where there is deep water recently upwelled. High chlorophyll levels can be found in the surroundings.

Author response: We agree with the physical mixing processes described by Reviewer #1 that would dilute the biomass (Chlorophyll concentrations) in the upwelling centres where the nutrient concentrations were highest. Our understanding of the Peruvian productivity paradox and the reason why it was mentioned here, seems to be a little different to the understanding of Reviewer #1. Figure 4 in Messie and Chavez (2015) shows how the potential new production, based on nutrient inputs via upwelling, are out of phase seasonally with estimates of primary production in the Peruvian Upwelling System. This seasonal mismatch is not indicated for any of the other three main Eastern Boundary Upwelling Systems (California, NW Africa, Benguela). We would suggest the following modification to line 67 to clarify this and to read (new text underlined) "... out of phase seasonally...". Corresponding changes made: Modified as suggested in line 66 (revised manuscript).

**Materials and methods**

Line 105. According to Figure 1, the range of 15 m was only at station A, at station B it was 5 m.

Author response: We thank the reviewer for picking up on this typing error. This should 40 - 55m and would be modified accordingly in the revised figure.

Corresponding changes made: This has been modified in Fig. 2 (formerly Fig. 1).

Line 109. ... collected from the mesocosms (M in Fig. 1)

Author response: Thank you for this suggestion. This would be modified accordingly in the revised manuscript. Corresponding changes made: This has been modified, please see line 120 in the revised manuscript.

Lines 129-131. The last sentence reads, "Both the surface (mesocosms) and treatment water (deep water) were filtered... However, the deep water added to the two biology treatments was unfiltered.

Author response: We can see that it is difficult to distinguish the two separate filtration steps used. In a modified manuscript we would use "screened" to refer to the gauze filtration to remove larger predators and "filtered" to refer to the 0.1um filtration used to remove microbes for both the inorganic and the organic treatments.

Corresponding changes made: These two filtration steps have been renamed, with the first filtration step (<64  $\mu$ m) as "screened" and thereafter just as deep or surface water (see lines 134-137) and the sterile filtration step of 0.1 $\mu$ m is referred to consistently now as filtered (lines 138-139). We hope that this makes the distinction between these two filtrations easier to follow.

Line 135 ... were set to the same two levels as in the organic...

Author response: Thank you for this suggestion. This would be modified accordingly in the revised manuscript. Corresponding changes made: This has been modified. See lines 146 – 147: "... were added to mimic the same inorganic nitrate and phosphate concentrations as in the high nitrate (HN) and low nitrate (LN) deep water."

Line 173. (picoeukaryotes, *nanophytoplankton*, small microphytoplankton, large microphytoplankton). It may be appropriate to add a few words here to inform that microphytoplankton is not well estimated by this technique, although it is recognized in the legend of figure S2.

Author response: It is correct that the distinction between the two microphytoplankton groups based on size and FL3 fluorescence in the cytogram has its limitations and was sometimes difficult to gate precisely. We suggest the following modification to acknowledge this in the revised manuscript: "Gating of the microphytoplankton groups based on size (small, large), was modified to the best fit for each sample, however, there is a source of uncertainty associated with this approach due to overlap in some samples between the groups (see Fig. S2 for two cytograms with identified groups)."

Corresponding changes made: See lines 201-203 in the revised manuscript.

Results

Initial conditions (Day 1)

Lines 256-257. If referring to all nutrients, Fig. 3B and C and Table S1 should be cited. If only nitrate is referred to, Fig. 3B should be cited.

Author response: Thank you for this suggestion. This would be modified accordingly in the revised manuscript from "nutrient" to "nitrate", citing just Fig. 3B in the first sentence.

Corresponding changes made: The results section has been substantially rewritten and the citation of the relevant figures was added where appropriate e.g. for Fig. 3B. Please see lines 279-283 in the revised manuscript.

Lines 262-265. Fig. 4E should be cited when discussing a254. For E2:E3 it should be Fig 4F. Add Table S1 to Fig 3F when LAP activity is discussed; the slightly higher activity is better seen in the table than in the figure. Author response: Thank you for picking up on these inconsistencies. We would modify these as suggested in a revised manuscript and thoroughly check all figure citations to ensure these are correct in the revised version.

**Corresponding changes made: Please see lines 284-293 in the revised manuscript.**

Lines 266-271. Table S1 should be mentioned when commenting about the phytoplankton community. The same table can be mentioned for Fv/Fm, Fig. 4D is the figure. Author response: We would add a reference to Table S1 and include the figure reference in a revised manuscript. See also our response to the previous comment. **Corresponding changes made: Please see line 298 in the revised manuscript.**

**Line 274. ...between Day 3 and 5. Better between Day 3 and 6 (Fig. 3D).**

It is difficult to follow the chlorophyll in this paragraph, it would be better to specify something else, for example: Peak Chl a concentrations of up to 12  $\mu$ g L-1 (HN organic) and ~6  $\mu$ g L-1 (LN inorganic and biology). According to figure 4A, there are differences between various treatments on this day 4.

Author response: We would add the specific reference to the treatments where the peak Chl a concentrations were observed as suggested by the reviewer to clarify where treatment differences were observed. This would then read "Peak Chl a concentrations of up to  $12 \mu g L^{-1}$  (HN) and ~6  $\mu g L^{-1}$  (LN) were observed on Day 4 (Fig. 2A). A significant treatment effect of nutrient concentration (HN - LN) was detected in the organic and inorganic treatments, and a significant treatment effect of biology (biology – organic) was detected in the LN treatment (Table S2a, b)."

**Corresponding changes made: This has been further modified from our suggested changes described previously. Please see lines 324-331 in the revised manuscript.**

Line 285. It is difficult to follow this about the ratio DIN drawdown to maximum Chla accumulation. This ratio was higher in LN only for the case of organic treatment (Fig. 4A). I think the next paragraph about higher recycling of N or highest N utilization efficiency under low nitrate needs further explanation. How this higher N recycling or N utilization efficiency deduces from a lower DIN ratio drawdown to Chla accumulation? It seems too risky to attribute these differences in the ratio only to N. Variations in the ratio may also be due to different cell concentrations of chlorophyll. Mixotrophic behavior can also affect this ratio. The ratio changes through changes in N, changes in chlorophyll, or in both. Here phytoplankton composition could provide additional information.

Author response: We thank the author for bringing up this point as we had viewed this observation with just one lens and it is very true that Chlorophyll a changes may also explain this result.

Although there are many limitations and uncertainties, we calculated the FL3 (chlorophyll) fluorescence per cell to see if any variations in cell chlorophyll content could be observed in the flow cytometry analyses, within the cell size range that is detected. This is an approximation of the chlorophyll content of chlorophyll-containing cells. Deviation between treatments did appear to emerge in the nutrient depleted period between Day 6 and 10 (see Fig. R1 below) and was likely driven by divergence in cell size between treatments that emerged around the same time (see line 293-296 and Fig. 4C in the manuscript). Highest mean chlorophyll fluorescence per cell was measured in the HN inorganic treatment and the lowest chlorophyll fluorescence per cell was measured in the LN biology treatment on Day 10 (Fig. R1).

Fig. R1: Relative cellular chlorophyll content estimated from flow cytometry data (FL3 fluorescence and cell counts) during the study.

We would incorporate the other possible explanations in the following suggested change to line 289 (new text underlined): "recycling of N or highest N utilisation efficiency under low nitrate in this treatment. Variations in the ratio may also indicate different cellular Chl a content or mixotrophic behaviour."

The description of the ANOVA output for the main effects and interaction effects, is a standard way of reporting the statistical data, however we can see this can be unclear to readers that are not so familiar with statistics. We would suggest retaining the ANOVA output in the text as is for lines 284-287, and then adding the following sentences thereafter to describe in plain words what this output means. This could be as follows: "This means that more Chl a was accumulated in the bloom per nitrate consumed in the low nitrate treatments compared to the high nitrate treatments. There was no significant difference however detected, neither between the treatments (inorganic, organic, biology), nor a combined effect (i.e. interaction) between nitrate concentration and treatment type. "

Corresponding changes made: In the revision of the results section, we realised that this was not an important result to support the discussion, therefore have removed the data, this section and the associated figure panels from the manuscript.

Lines 291-292. The last sentence indicating that the initial concentration of DIN was 3 times higher in HN than in LN can be deleted. It was reported at the beginning of the results. Author response: *This sentence would be deleted in the revised manuscript.* Corresponding changes made: Sentence deleted.

Line 322-323. I understand the association between higher silicate drawdown and higher chlorophyll concentration, but not with nanoplankton abundance. There is no information on the species that are in the nanoplankton fraction. On the other hand, the increase in chlorophyll could well occur in micro diatoms. Maybe the sentence could write like this:

The highest Si(OH)4 and phosphate drawdown, and consequently Chl a concentration was observed in one replicate. This replicate also showed highest nanophytoplankton abundances (Fig. 5B).

Author response: This sentence would be modified as suggested by the reviewer, including the reference to Fig. 5B.

Corresponding changes made: We have changed the results section here to read as follows (lines 377-381): "There was an increase in average cell size in the high nitrate "biology" incubations (Fig. 4A), in addition to particularly high post-bloom silicate consumption in the high nitrate "biology" incubations as well marked differences in silicate uptake between the replicates in the high nitrate "biology" incubations (see large error bars in Fig. 3D). This high variability was driven by the response of one high nitrate "biology" replicate that

also had the highest phosphate drawdown, and post-bloom Chl a concentration and also had highest nanophytoplankton abundances (Fig. 5E)."

Fig. 5. I think the symbols on the panels do not correspond to the ones on the labels, where they are all circles. Author response: We apologise that part of the figure legend for the symbols is missing, and thank the reviewer for picking this up. There are four different symbols (circle, square, triangle, diamond) used to distinguish the four replicates. This information would be added to the figure in the revised manuscript.

Corresponding changes made: A symbol legend for the replicates A-D has been added below Fig. 5.

**Discussion**

Line 370-372. I think this sentence about bottom-up and grazing control is missing something. Author response: Yes, a verb is missing. The sentence should read "... high nitrate inorganic treatments suggest a primarily bottom-up driven food web response ...".

Corresponding changes made: Please see line 395 in the revised manuscript.

Line 420-422. Silicic acid consumption could well have occurred by micro-sized diatoms. It is difficult to conceive that all or nearly all of the nanophytoplankton were diatoms. Usually, there are many flagellates in this fraction.

Author response: We agree with both of these points: that some silicic acid could have been consumed by larger (micro-sized) diatoms and that flagellates were likely abundant in the nanophytoplankton group. However, the divergent response in the nanoplankton size class and silicate drawdown in the one replicate, suggests it was a silicifying species that consumed a lot of silicate. This could have been a silicifying nanoflagellate but as these are usually in the micro size range >20  $\mu$ m (Hernández-Becerril and Bravo-Sierra, 2001) and due to the magnitude of silicate drawdown, we considered this more likely to be a diatom species. The interesting point we find here, is that the divergent biological response had an impact on the nutrient concentrations, even if we cannot precisely attribute this to a particular species. We hope that this outcome is clearly presented and is understandable in the manuscript. We would suggest the following modification to line 421-422 as follows (new text underlined): "... likely diatoms based on the magnitude of dissolved silicic acid consumption."

Lines 429-430. Diatoms were not analyzed and, therefore, it cannot be confirmed that the different behavior of the two treatments was due to the different response of the diatoms and the different seed population. What can be said is that the different behavior of the two treatments could be attributed to a different response of the diatoms and probably also to differences in the seed population.

Author response: We would change "diatom community" to "silicifying phytoplankton" in line 428/9 accordingly, to more broadly refer to silicic acid consuming phytoplankton.

Corresponding changes made: We have used the term "silicifying phytoplankton" instead of diatoms to address this issue raised by the Referee. See e.g. lines 441 and 458 in the revised manuscript.

Line 490-491. The highest silicate uptake only occurred in a biology treatment, in the HN biology. In the LN biology it did not occur (Fig. 3D).

Author response: Yes this is correct, and this is acknowledged in line 491 ("within a given deepwater"). Corresponding changes made: No change made.

**Author response to Referee #2 comments (Citation: https://doi.org/10.5194/bg-2022-44-RC2) including a list of all relevant changes made in the manuscript corresponding to each referee comments.**

We thank the Referee for their constructive comments and suggestions and respond to these point-by-point below. Our comments are presented in *italics*.

Additional note: We added the details of any changes made to the manuscript below our initial comments in **bold** ("Corresponding changes made"). All line numbers in our responses refer to the revised manuscript (no track-changes).

**General comments**

This ms describes the time evolution of nutrients, chlorophyll, flow cytometric groups, chromophoric dissolved organic matter in experimental 15L bags during 10 days after mixing surface sea water with an equivalent volume of 'deep' water under different conditions (nutrient rich, nutrient poor, filtered (with natural organic matter and nutrients but no organisms) or not (with natural nutrients, organic matter and deep microorganisms). A nutrient control was set by mixing surface sea water with filtered surface sea water and adding nitrate and phosphate. The design of the experiment was set to explore not only the effect of nutrients on the sea surface nutrient consumption and phytoplankton dynamics and composition under the influence of upwelling episodes, but also that of nutrients + organic matter, and that of nutrients + organic matter + natural deep communities upwelled from the 'deep waters'.

If the objectives of the experiment are clear, a huge work and many parameters shown, an interesting introduction and discussion, the design and the results are very complicated to follow. I suggest to the authors to simplify denominations/definitions of the different combinations in the bags and to rewrite the m&M and the result section to give it clearer. In addition, I was left a bit frustrated as:

For a biogeochemist point of view, we have no information on ammonium concentrations. Regenerations sources could have been high in the bags, particularly considering the high concentrations of nitrate added: around 8 and 3  $\mu$ M in HN and LN, respectively. This information should have been pertinent to state nitrogen limitation status and examine N/P ratios (and for this reason, you should use the term NOx instead of DIN for the whole text, as it refers only to the sum nitrite + nitrate)

For a microbiologist point of view, again, we have only partial information:

First on the microphytoplankton response: There is discussion on a diatom response, but we have not any information on the taxomomic composition of diatom communities, even using proxies that could be have been given for instance by size fractionation of chlorophyll. Indeed, flow cytometry analysis only allows small size class to be counted. There was only few information given on a "chain group", on a "small microphytoplankton group", and on a "large phytoplankton group" which was not statistically represented in

the flow cytometry analysis (Fig. S2), and furthermore the time evolution of abundances in these groups is not plotted (just initial conditions, table S1) or statistics (table 2).

Second, there is no information on heterotrophs. There is discussion on the potential role of heterotrophic bacteria on the degradation of organic matter, and on the the phytoplankton regulating process during its decay period. But not any data is available on heterotrophic bacterial abundances and/or on the grazer community composition, or virus abundances.

The choice of deep samples at 90-105 m at station A for HN experiment, and 40-45m at station B for LN experiment is crucial to compare experiments. Clearly the vertical distributions of physical properties, nutrients, abundances of flow cytometric groups and organic matter properties at these 2 stations, could have been helpful for the readers, particularly those not familiar with the biogeochemical context in this area, to situate these "deep" water masses in the frame of depths of nutriclines, OMZ, deep chlorophyll maxima, euphotic zone depth etc... Were these depths taken out of the euphotic zone? out of the OMZ?, Were they constituted only by heterotrophs or were there also some phytoplankton?

And Lastly, why taking waters from the mesocosm experiment instead to in situ surface waters close to the mesocosm? Why day 20?

For all these reasons, I would recommend the publication of the ms with major revisisons.

Author response: Both reviewers have commented that the Materials and Methods and Results sections would need some revising to improve clarity of this complicated set up. We would provide significant modifications to the text in these sections and work on simplifying the treatment definitions and simplifying the text. We would also change "DIN" in the text to "NOx" as suggested. The depth range of 40-45m should read 40-55m and was an unfortunate typing error for Station B, which we will correct during manuscript revisions.

Corresponding changes made: We have made significant modifications to the Methods and the Results sections which we think has improved the clarity of the set up and experimental design. We also changed the experimental design figure to better represent the treatments and more clearly distinguish the two nutrient levels used and the three deep water components isolated, and also more clearly indicate the treatment names. We also added a section to the Methods to separate the experimental design information (Section 2.1) from the water collection, incubation set up and sampling procedures (now Section 2.2). Use of DIN and NOx has been checked and made consistent throughout the manuscript and Supplementary Material as NOx, including in relevant nutrient ratios.

During the deepwater collection, we went out along the Callao transect during sampling by Instituto del Mar del Perú (IMARPE). This transect extends off the coast along 12°S and is regularly sampled (see e.g. Graco et al. (2017) or also the Graco et al. (2019) as cited in line 97 for background hydrographic data and biogeochemical context). We took a CTD profile at each station and looked at the oxygen, pH and hydrogen sulphide (H2S) profiles to indicate in which depth range water was still oxygenated and therefore should still contain some nutrients, in particular nitrate, and no H2S as this is toxic for plankton. We then selected depths above any anoxic layer and as deep as possible to ensure high(er) nitrate concentrations, which was a central factor for the design of our study.

**Corresponding changes made: Please see lines 108-114 in the revised manuscript for this information.**

Regeneration of nutrients, in particular nitrogen, is very likely in this N-limited system but unfortunately no ammonium concentrations could be measured in this set-up due to the high risk of sample contamination. We respond to this specific comment in more detail below in responses to specific comments, including how we would address it when revising the manuscript.

Size fractionated Chl a would have been one way to analyse the size structure of the phytoplankton community. Indeed, flow cytometry analyses can give a proxy for cell size that can be even more highly resolved than the standard <2um, >20 um fractionation carried out in filtrations, although as correctly mentioned, this is limited by the size range the flow cytometer can detect. We were also quite limited in sample volume for this 10-day long study, hence Chl a samples were only taken every 2nd sampling day, whereas flow cytometry only requires ~2mL and could be taken more frequently. Hence, we were interested in the temporal dynamics and placed priority on this as we were not sure how rapidly the plankton community would respond to the nutrient addition in this particular set-up. Please also see our response to a similar comment by Reviewer #1.

It is an oversight that the heterotrophic community was not analysed in greater depth as this would give better insight into regenerative production and phytoplankton bloom dynamics. Virus abundances are notoriously difficult to analyse and we did not have the means to do this in this particular study, but of course would be an interesting dynamic to probe in future studies.

We wanted to use mesocosm water rather than surface water because the water masses inside the mesocosms are comparatively well characterised and experience less mixing and no tidal movements, unlike outside the mesocosms. The initial aim was to link the results of this incubation study to the mesocosm responses, but this was more challenging in hindsight, nevertheless determined the experimental design initially. Day 20 was selected because we had ship time to go out and collect the deepwater in the days prior, because the initial collection did not work. This is perhaps a benefit as around Day 20 the nutrients were depleted but primary producers were still relatively abundant so that our treatments could better control the nutrient concentrations.

Corresponding changes made: When rechecking all DIN data during revisions, we realised that ammonium concentrations were measured but the concentrations were unreliable and so were not included in the

manuscript, or the DIN:DIP or DIN concentrations reported previously. Unfortunately they were included in the DIN:DIP drawdown calculation, hence we have now recalculated this as ΔNOx:ΔDIP, and redone the statistical analyses to correct for this error. This did not change the statistical output very much. The corrected output is reported in Table S3a and S3b in the supplementary material and necessary corrections in the manuscript text (e.g. line 318 in the revised manuscript) have been made accordingly.

**Specific comments**

Line 109. Specify even of you refer to Bach et al what was the status of phytoplankton on the day 20 of the mesocosm study: steady state? exponentially growing? decaying?

Author response: We would add the following description to provide information on the phytoplankton status in the collected surface water to Line 112: "The mesocosm plankton community was in a post-bloom phase where inorganic nitrogen was low and a sub-surface Chl maximum had developed between 5-15m depth." Corresponding changes made: Please see lines 123-125 in the revised manuscript for this addition.

Line 119. The authors should moderate this sentence "... no TM or DOM measurements were made... these were assumed to be different...."

Author response: It is unclear what specific part the reviewer suggests moderation of in this sentence. As some treatment differences were detected between the inorganic and the organic treatments for the HN deepwater, but not for the LN deepwater (see e.g. Table S5c), we believe that this corroborates this statement. As this is presented in the results section and supplementary material, we do not consider it relevant to add this information to this early part of the material and methods.

Corresponding changes made: No change made.

Figure 2. station A and station B should be removed in the lines "inorganic", somewhere it should be drawn that nitrates and phosphate were added. In the legend for memory it should be reminded that "unfiltered" is in fact < 64  $\mu$ m and the "filtered" is < 0.1  $\mu$ m.

Author response: We can see that it is difficult to distinguish the two separate filtration steps used, as both reviewers have highlighted this point. In a modified manuscript we would suggest use "screened" to refer to the gauze filtration to remove larger predators and "filtered" to refer to the 0.1um filtration used to remove microbes for both the inorganic and the organic treatments. We will clearly describe this in the manuscript and consistently use this terminology throughout. We would also revise Table 2 (please see our response to this aspect in "General Comments" above).

Corresponding changes made: We have redrawn Fig. 2 (now Fig. 1) and included more information which we hope clarifies these steps. We also made the distinction between the filtration step and the initial screening step clearer (see lines 134-139 in the revised manuscript).

Line 135 for the "inorganic "treatment why not also adding Silicates to get similar changes for N/Si ratios? Author response: This study was intended as an auxiliary experiment to the larger mesocosm study (see description in Bach et al. 2020, BG), where deepwater with two different N:P ratios were added. Hence, we selected the nutrient treatments based on N and P only also for this study. Adding an additional N:Si treatment would have significantly increased the number of incubation bottles needed and would not have been manageable workload within the framework of this study or possible within the limitations of the experiment set-up in Peru. Nevertheless N:Si ratios would definitely be an additional interesting factor to investigate in future experiments, as it is also likely important in selecting for particular phytoplankton groups. Corresponding changes made: No change made

Line 138, 140. Be more precise and everywhere add the porosity of the filtration. *Author response: Please see our response to the reviewer comment on Figure 2 above.* Corresponding changes made: Please see our response above regarding distinction of the screening and filtration steps. Line 172. As the samples were not fixed, how were stored samples between sampling and analysis? was there a long delay between the first and the last sample analyzed?

Author response: Sampling was rapid and on-shore so samples were taken to the lab within 1 hour of sampling and measured within 6 hours of sampling from the incubators. Samples for flow cytometry were stored cooled in the dark in the cool box until analysis.

**Corresponding changes made: This information has been added to lines 166-167 and 191-192 in the revised manuscript.**

Line171-180. The authors should add more details on the set up of the flow cytometer (and/or on the legend of Figure S2): debit and duration of the analysis, i.e. the volume analyzed), also did you add beads to set limits between the different size classes and how this differentiation was done. Add information on wavelengths of the different windows (FL3A, FL2H, FL4H) that could be indicated on the legend of the Figure S2. In this figure some cytograms were excluding some populations and other not? explain more in the legend. As demo, a plot FL2 / FL3 would have been useful too.

Author response: We will add additional information to the methods section to better describe the flow cytometer set-up and analysis in accordance with the Reviewer suggestions here. Each sample was measured over ~10 minutes per sample on fast flow rate (~66  $\mu$ L per minute) to measure a total volume of 650  $\mu$ L. No beads were added, instead sizes were determined via sequential size fractionations with polycarbonate filters of different poresize as described in Veldhuis and Kraay (2000). We used the wavelengths excitation/emission of FL3 = 488/670, FL2 = 488/585, and FL4 = 640/670. We considered all populations for the quantitative analysis but of course not all populations are necessarily shown on all plots. For gating, some identification was needed on specific fluorescence channels (e.g. Synechoccocus on FL2) and these are then excluded from the other plot (e.g. FL3 vs. FSC) to avoid overlap with the other populations. This is a standard gating procedure. We will also add a plot of FL2/FL3 to the Supplementary Material, as is provided already for the size-fluorescence cytograms in Fig. S2.

**Corresponding changes made: Please see lines 190-209 in the revised manuscript and Fig. S2 in the revised Supplementary Material.**

Could you really consider counts of micro-II as significant? we just have initial conditions in terms of percentage on table S1, but no idea of any abundance,

Author response: The abundances of the micro II (large microphytoplankton) were much smaller than for other groups. These ranged between 0 and 11 counts per sample (analysed volume =  $100 \mu$ L) but were included due to their larger size and contribution to chlorophyll fluorescence signal. We focussed our attention in the discussion however on more dominant groups where we also considered the underlying data to be more robust. Corresponding changes made: Please see lines 199-203 as well as lines 300-304 for addition of this information into the methods and results sections of the revised manuscript.

For a paper dealing on the potential effect of seeding microorganisms with upwelled waters, I find rather strange to get so much groups counted with the cytometer and having finally only a plot of the evolution of *Synechococcus* and nanophytoplankton.

**Were Prochlorococcus abundances detectable?**

Author response: In the manuscript we focused on the key outcomes and main effects observed, hence the focus on Synechococcus and nanophytoplankton. As there was little impact of treatment on the other groups gated, we highlighted this outcome, even though we of course measured and gated other groups. We will add these figures to the Supplementary Material to show how these groups varied by time and treatment.

Prochlorococcus was not detectable in this study. From experiences in other incubation and mesocosm studies, Prochlorococcus usually does not survive in the bottles or other enclosed spaces, even if they are present in the initial community.

Corresponding changes made: No change made.

Line 190. Were all the FV/Fm measurements done on the same time of the day? Author response: Yes, they were carried out at the same time. After sampling, the samples were adapted to dark conditions during 20 mins prior to measuring F0, Fm and Fv. Corresponding changes made: No change made.

Paragraph 2.3 Is there any information on the distribution of heterotrophs? heterotrophic bacteria? Heterotrophic nanoflagellates? ciliates? the filtration on < 64  $\mu$ m could have a cascading high effect on ciliates, which becomes the top predator in the bags.

Author response: We have presented all data collected during the study and unfortunately there is no further information on heterotrophic organisms. This is definitely a component of the food web that would be interesting to probe in more depth in future experiments, as rightly mentioned, these smaller grazers (<64um) could have profited from the relief of grazing pressure on themselves and may also display mixotrophic behaviour.

**Corresponding changes made: No change made.**

Line 229. Please explain better for non-specialists: "the contrast matrix... the organic treatment was used as the control for the linear mixed model analysis"

Author response: The following sentence was meant to explain the importance of this contrast matrix and would be modified to make this clearer e.g.: "This contrast matrix hence means that reported model significance refers to the difference between organic vs. biology and organic vs. inorganic to distinguish the biological treatment effect and the organic/trace metal effect.

Corresponding changes made: Change made as described to line 251-253 in the revised manuscript.

**Line 256. refer also to Table S1**

Author response: We would add this to the reference to Fig. 3 as follows: "... (Fig. 3, Table S1) ..." Corresponding changes made: This reference has been added along with many others during revision of the results section.

Line 259. refer a254 with Figure 4e, and on line 262 E2:E3 ratio with fig 4f Author response: We would add this information as suggested by the reviewer to line 259 to read: "... CDOM measurements (a253, Fig. 4E) ..." and to line 262 to read "... DOM molecular weight (E2:E3, Fig. 4F)." Corresponding changes made: This reference has been added along with many others during revision of the results section.

Line 264. "surface water.... nitrogen depletion". Initial concentrations in NOx in "inorganic" (2.07) versus those in organic LN (2.49 and 3.17 (organic LN) were not so different and where all evolving the same way considering nitrate (Figure 3b) or LAP activity (Figure 3F)

Author response: We are not entirely sure what would need modification here. As the reviewer points out, the NOx concentrations (inorganic) were very similar between the treatments. This was deliberate so that any contribution of the organic nutrient component added via the deepwater treatments could be distinguished, if this was a significant effect. Lack of significant difference in LAP activity also indicates that organic nutrients (nitrogen here) did not play such an important role in fuelling production in the nitrogen limited plankton community here.

**Corresponding changes made: No change made.**

Line 266. please explain in the Material and method in 2.3 section clearly how is calculated the "relative contribution of each group to chlorophyll a fluorescence in flow cytometric analysis" Also if it is this parameter used on table S1 when reporting percentages of the different flow cytometric groups, and not simply relative abundances, it should be explained in the legend.

Author response: We would add this information to the Materials and Methods section in the revised manuscript and ensure this is explained in the Table S1 caption, as this data does underlie the percentages reported in Table S1.

Corresponding changes made: This information has been added to lines 207-208 in the revised manuscript and to the Table S1 caption.

Line 268. Change formulation "R statistic" here and in Table 2 Not the software, but the type of test should be indicated.

Author response: The "R statistic" here is not referring to the software but to the output from the statistical tests (ANOSIM + SIMPER) that is described in the methods section in Section 2.6. **Corresponding changes made: No change made.**

Line 273. It is not visible in Fig 3B plot, were the limits of detection for nitrate (0.123  $\mu$ m as stated in the

methods) reached in all experiments? Author response: Yes, all replicate bottles had NOx concentrations below the detection limit on Day 4 with some bottles below detection on Day 3. This data is now openly accessible on the PANGAEA database: https://doi.org/10.1594/PANGAEA.941138

Corresponding changes made: The link to the data in the PANGAEA data repository has also been added to the manuscript in line 532 under "Data availability".

Line 282. Is there an error here? I would rather write this sentence like this: "DIP is more consumed relative to DIN in LN treatments compared to HN treatments ...."

Author response: Yes, this is correct and we thank the reviewer for bringing this up. We would modify this sentence accordingly in the revised manuscript.

Corresponding changes made: This change has been made to line 347-350 in the revised manuscript as follows: "Treatment related differences in nutrient uptake ratio were also detected in the post-bloom phase with over 50% higher consumption of DIP compared to NOx in low nitrate than in high nitrate incubations by the end of the 10-day long study period ( $\Delta$ NOx: $\Delta$ DIP on Day 10: high nitrate = 9.32 ± 1.68, low nitrate = 5.43 ± 1.61, mean ± s.d., n = 6, see also Fig. 3E)."

Line 283-284. Is there an error here? rather it should be 9.82 for HN and 6.25 for LN? Author response: Yes, this is correct and we thank the reviewer for bringing this up this error. We would modify this sentence accordingly in the revised manuscript.

Corresponding changes made: This correction has been made to line 349-350 in the revised manuscript.

Line 285. "... where initial N was lowest". No, initial N was the lowest in the inorganic treatment. Author response: We can see that the treatment naming is difficult to follow and we will work on this in a revised manuscript. To clarify this specific result, here we were referring to the nutrient status (high vs. low nitrate) without any reference to any of the three treatments (inorganic, organic, biology).

Corresponding changes made: The relevant sentence here has been deleted during revision of the results section.

Line 289. "higher recycling...". Ammonium concentrations were not available? Probably like DIP it was produced by regeneration between days 6 and 8, see lines 325-326

Author response: Ammonium concentrations were not measured in the nutrient analyses. Regeneration of nutrients is likely, as we suggest in lines 325-326, as the detection limits for DIP are much lower than for NOx. Also, the incubations were limited in NOx, rather than DIP, any regenerated N would have been rapidly assimilated and hence would not accumulate and remain undetectable in any of the NOx pools, or in ammonium if it had been analysed.

Corresponding changes made: No change made.

Line 289. ".. or highest N utilization efficiency under low nitrate" There is another hypothesis, a higher topdown control of phytoplankton by grazers under high nitrate.

Author response: We address this more in depth in our response to the Reviewer's feedback on line 372 (see below).

**Corresponding changes made: No change made – see response to feedback on line 372 below.**

Lines 291-292. This sentence on initial nitrate nitrite concentrations should be cited at the beginning of section 3.1

Author response: The initial NOx (nitrate + nitrite) concentrations are already mentioned at the beginning of Section 3.1 (line 256-257).

**Corresponding changes made: No change made.**

Figure 3. Why plotting silicate drawdown when absolute concentrations are plotted for Chl, DIN and DIP? For plots based on 'deltas' like figure 3D and 3E the authors should explain in the legend if the difference is always made with T1 concentrations

Author response: Silicate was never depleted and the point of the figure was to show the differences in silicate consumption, despite nitrate/NOx depletion. We would add information to the legend to indicate the silicate drawdown was calculated from the T1 concentrations.

Corresponding changes made: We added an equation and a description of how this was calculated to the methods section (lines 177-182) and added Equation (2). We also added information to the legend of Fig. 3 to read "... silicate drawdown relative to initial concentrations on Day 1 (Eqn. 2)".

Figure 4A. For the legend, indicate how DIN was calculated. Was it DIN at T1 minus DIN at the time of max chlorophyll, i.e. T4 for all samples except T3 for LN organic?

Author response: We will add a reference to Eqn. 1 in the legend to indicate how this was calculated. This ratio was calculated for each replicate individually (see line 247) and the day of nutrient depletion ( $DIN_{min}$ ) and Chl  $a(_{max})$  may therefore differ.

Corresponding changes made: These panels have been removed in revision of the manuscript.

Figure 4B. Again it is unclear if the difference is made as concentrations at day 6 minus concentrations at day 10 of these box plots are simply the means of the data presented figure 3E for the period T6 to T10. Be clearer in the legend. Is seems that here are presented the distribution of the 20 data (T6 to T10 time points x quadruplicates bags).

Author response: We will modify the figure legend to clarify this as the mean of daily calculated values between Days 6-10 to read as follows (new text underlined): "...B) mean relative drawdown in nitrate to phosphate during the post-bloom period (mean of daily calculated  $\Delta DIN:\Delta DIP$  on Days 6-10), ..."

Corresponding changes made: These panels have been removed in revision of the manuscript.

**Line 324. Refer to fig 5E**

Author response: We will add this figure reference accordingly in a revised manuscript. Corresponding changes made: These figure reference (Fig. 5D-F) has been added to line 353 in the revised manuscript.

Line 330. The last sentence with infos on initial conditions should be cited in section 3.1 Author response: We will shift this information to Section 3.1 in a revised manuscript. Corresponding changes made: This information can now be found in Section 3.1 (lines 279-281).

Table 2. Modify "R statistic", indicate in the legend what were the bloom and post bloom periods considered for the tests. For the relative contribution, in percentages, I don't understand to what they refer, as the sum of contribution of each group does not make 100%.

Author response: We will add information to the legend on the definition of the bloom and post-bloom periods.

As described in our response above, the "R statistic" here is not referring to the software but to the output from the statistical tests (ANOSIM + SIMPER) that is described in the methods section in Section 2.6 and is correct to report as an R statistic. The contribution reported in the table is the contribution to the detected treatment differences in the SIMPER Posthoc test. This is described in the legend for Table 2.

Corresponding changes made: We have added the following description to the caption for Table 2: "Bloom phase is defined here as Day 3 and Day 4, and Post-bloom is defined as Day 8 to Day 10."

**Line 369. It is up to 12 $\mu$ g/l as seen from the figure 3A**

Author response: The maximum Chl a concentration was  $12 \ \mu g \ L^{-1}$ , as the reviewer correctly points out. However, here we refer to the high variability observed in maximum Chl a concentrations between the three treatments (organic, inorganic and biology) for the high nitrate incubations. Hence the  $6 \ \mu g \ L^{-1}$  difference between the treatments we refer to in Line 369 is in our opinion correct here.

Corresponding changes made: We have significantly revised the text in the results section and hope that this has become clearer. Please see lines 324-325 in the revised manuscript.

Line 372. "... than any impact of grazing": But some grazers were present in the surface water taken in the minicosms. Furthermore, this surface water was filtered through 64 µm, and consequently with no top predators, all microzooplankton (heterotrophic ciliates) could have been rapidly growing in response to the increase of pico nano and small microphytoplankton. Note also that his sentence lines 370-372 has no verb. Author response: We agree that other groups that could graze on phytoplankton in different size classes may have been growing rapidly in the incubations and can see that this statement is a little unclear and would remove the reference to grazing impact. Hence, we would revise this section to read as follows: "Sharper bloom biomass development in the filtered high nitrate organic and high nitrate inorganic treatments suggests a primarily bottom-up driven food web response to nutrient addition. Bloom development in high nitrate biology was more muted as nutrient competition within the plankton community (e.g. with heterotrophic bacteria) was likely higher, due to the lack of organism dilution compared to the filtered organic/inorganic nutrient treatments. Alternatively, this muted biomass development could suggest an increase in grazing pressure via potential introduction of microzooplankton grazers (<64µm) in the addition of unfiltered deep water. Hence, higher retention of Chl a post-bloom in this treatment, suggests potentially longer sustained periods of productive biomass when deep water plankton are added concurrently with upwelled nutrients. The precise mechanism(s) underlying this response requires however further detailed elucidation."

Corresponding changes made: This information has been added to lines 394-402 in the revised manuscript.

Line 372-374. Rather, I would imagine than heterotrophic bacteria would find more favorable growth conditions with surface water mixed with deep filtered sea water, as the surface heterotrophic bacteria are diluted in deep water by a factor 2 as well as their grazers, and thus have less predatory control, together with more access to nutrients and DOM provided by the deep waters.

Is there any information on abundances of heterotrophs? heterotrophic bacteria? flagellates? ciliates? Author response: We agree that the surface heterotrophic bacteria would also benefit from the dilution in the sterile filtered deepwater but as we didn't measure their abundances, this is difficult to quantify. In line 374, we were referring to the increased competition in the unfiltered biology treatment where this dilution, and potential increase in relative inorganic/organic nutrient concentration, did not occur. Corresponding changes made: No change made.

Line 385. The noticeable net increase of DIP in experiments between T6 and T8 suggests that ammonia could have been also regenerated through grazing processes during that period. This increase of DIP about 0.1 to 0.3  $\mu$ mole/I, based on a N/P ratio of 16, could signify that as much as 1.6 to 4.8  $\mu$ M of ammonia could have been regenerated, even based on a delta DIN/delta DIP of about 6, this give up to about 2  $\mu$ mole/I ammonia regenerated.

Author response: Yes, we agree that ammonia regeneration for example due to grazing by smaller grazers still present in the experiment is also likely but it would have been rapidly assimilated in the N-limited plankton

community and hence remain undetected in the dissolved nutrient pools. Nevertheless, this may have been visible in the Chl a biomass as we suggest in line 384. This regeneration of N via ammonia could have helped to sustain Chl a biomass at higher levels during the post-bloom period. Unfortunately, limitations on the size of the incubations meant we could not carry out grazing assays that require large volumes (>5L).

**Corresponding changes made: No change made.**

Line 393. The authors should cite the initial DIN/DIP ratios here, and write that DIP was never depleted in the experiments.

Author response: We would modify the sentence here to read (new text is underlined): "Our results also indicate that up to over 50% more phosphate was consumed per mol of DIN in the low nitrate treatment (mean initial DIN:DIP =  $1.21 \pm 0.24$ ) with higher initial excess phosphate, than with high nitrate (mean initial DIN:DIP =  $3.92 \pm 0.32$ ). Phosphate was never depleted in this study. "

**Corresponding changes made: Please see lines 419-420 in the revised manuscript that also now uses NOx instead of DIN.**

Line 396. Without ammonia measurements, it is difficult to speculate on N regeneration. However, if abundances of heterotrophic prokaryotes are available, I suggest to calculate per cell LAP activities. Author response: Unfortunately, heterotrophic prokaryotes were not distinguished in this study and so we cannot calculate per cell LAP activity.

Corresponding changes made: No change made.

Line 401 "LAP was higher...". Before comparison with other studies the concentration of leu-AMC used by other authors should be verified as it influences rates. The concentration added here (500  $\mu$ M) is high. Mabmig et al used 200  $\mu$ M.

Author response: Yes, this is correct, and we suggest the following modification of this section to include this (new text underlined): "The LAP activity was one to two orders of magnitude higher than most literature values. For example, in a study from the same region but further from shore, the LAP activity was 20-65 nmol AMC L-1 h-1 in natural communities (Maßmig et al., 2020). Partly, the high LAP activity in this study could be due to the high concentration of substrate we used (500  $\mu$ mol L-1 leu-AMC), which aimed to measure maximum hydrolysis rates. However, this cannot be the only reason for the high values. For comparison, we used only 2.5 times higher substrate concentration compared with Maßmig et al. (2020). The high LAP activity and close relationship with fresh, labile organic matter production suggests that LAP was produced to support bacterial production above the oxycline (Loginova et al., 2019), rather than compensating for higher N–limitation in the low nitrate treatment. "

Corresponding changes made: Please see lines 429-434 in the revised manuscript that includes some minor modifications to our previously suggested changes.

Line 428. "Irradiance levels increased upon incubation". Were the levels of irradiance in bags higher than in the surface mesocosms?

Author response: The blue foil used reduced the incoming irradiance to ~25% which corresponded to equivalent PAR at ~ 2-3m deep (25% light intensity) and we aimed to use representative light conditions in the incubation study. However, the water in the mesocosms was in general mixed within the upper mixed layer of ~ 5-10m, meaning that the average PAR was probably below 25% light intensity in this mixed layer. So while the light level in the incubation was in good agreement with the depth where the water was collected from, the reduction in vertical mixing upon incubation would have increased the amount of incoming irradiance and likely increased the amount of PAR available to phytoplankton.

Corresponding changes made: No change made.

Line 440 "viral presence". Because the authors made a 0.1  $\mu$ m filtrations the ratio viruses to their host is very high in initial conditions., could it be in the favor of viral lysis?

Author response: The reviewer correctly notes that we filtered the deepwater added to remove as many particles such as viruses from the deepwater as possible but of course some smaller viruses may have been retained in the sterile filtered water (inorganic and organic treatments). The whole size spectrum of viruses (<64 µm) present in the unfiltered biology treatment. Viral lysis may therefore have been favoured in these filtered treatments due to the higher concentration of smaller viruses. This would be an alternative argument to the one we use but we have no data to support this.

Corresponding changes made: No change made.

Line 433. "... rather then the manipulated deep water". It would have been interesting to have initial compositions of populations included in the two "deep waters" used in this experiment.

Author response: We took subsamples from the seawater added and analysed these using flow cytometry directly after mixing but not of the deepwaters themselves, as we were interested in the treatment differences. This drove our sampling strategy. While we agree it would have been interesting to better characterise the initial populations of the deepwaters specifically, our aim was to see if a different water source was at all an important driver of phytoplankton communities and considered characterisation of the individual treatments sufficient when carrying out the experiment. In future studies, we would strongly consider genetic sampling to better characterise the microbial communities present (i.e. 16S and 18S) and even transcriptomic analyses to understand differences in nutrient utilisation.

Corresponding changes made: No change made.

Line 455. "...and higher post-bloom Chla concentrations were sustained in this treatment". Yes, but in the "inorganic" too, so the source of the variability is not only due to the variability of responses of the seeded communities, those of surface too.

Author response: Yes, we agree that there were similar mean ChI a concentrations post-bloom in both the biology and inorganic treatments (see e.g. Fig. 3A). Nevertheless, the heterogeneity, or variability, between replicates was much higher in the biology treatment, evident in the error bars in Fig. 3A, 3D and 4A. We would suggest the following changes to line 455 (new text underlined): "Over 2.4 µmol L-1 more silicate was consumed after nitrate was exhausted and higher post-bloom ChI a concentrations and nanoplankton abundances were sustained in the one HN biology replicate compared to the three other replicates."

Corresponding changes made: Change made as suggested to line 486-488 in the revised manuscript.

Line 480. Sentence unclear, does the term "that" refer to physical factors? If yes do you discuss about the horizontal mixing by showing the example of tidal mixing? If yes write it.

Author response: We would suggest the following revision with the modified text underlined: "In the ocean, physiological and ecological drivers (e.g. growth rates, transcriptional response times, mutualisms, symbioses, Sect. 4.2) would act in addition to other physical factors that regulate plankton biomass accumulation and succession in the surface waters following upwelling e.g. seed community abundance present in subsurface waters (Seegers et al., 2015). Such physical factors, such as dilution, mixing and horizontal transport of water masses via regular tidal transport onshore (Stauffer et al., 2020), could not be included in this experimental set-up."

Corresponding changes made: Please see lines 510-514 for the relevant changes made in the revised manuscript.

---

## Author Response (AR2)

**Editor comments to the author**:

Dear authors,

both reviewers have rechecked your proposal and reviewer #1 suggested technical corrections whereas reviewer #2 suggested minor corrections:

"Although the manuscript has improved substantially, there are two sections in material and methods that I think need further improvement. I am referring to section 2.1 Experimental design and section 2.2 Water collection, incubation setup and sampling; specifically section 2.1 on experimental design.

In my opinion, the modifications introduced in this new version of the manuscript do not help to understand the experimental design. Once screened water and filtered water are clearly differentiated, the whole process becomes clearer. I do not think it is necessary to name the screened water as unfiltered water in any case, since all collected water was screened to eliminate predators. It is important to make very clear that equal parts of mesocosms water (50% filtered and 50% unfiltered) were used in the inorganic treatments, to which nutrients were added at the level of the deep water in the corresponding HN and LN treatments. In the previous version, it was already clear that 50% of filtered water from stations A and B (HN and LN treatments) was added the organic treatments.

Honestly, I think this part of the manuscript needs improvement. The experiment is complex and the reader must easily understand what has been made, in order to also be able to follow and understand the results."

Thus, it is required to take this into account in your revised version of the ms!

**Author response to the Editor**

**Dear Editor,**

**We thank you for your feedback as well as the referee comments. This guidance has certainly refined and improved the methods and results section as addressed these comments and describe these changes below the relevant comment below.**

**The experimental design is complex and we carefully considered how to best present this information and have modified this section considerably and also consulted colleagues who did not participate in this study and are not co-authors to clarify which aspects were difficult to understand and needed revising. One referee seemed very satisfied with the manuscript after the last revisions as well as with how we responded to referee remarks, and one referee was less so. The order of presentation is what we understand as the main criticism, rather than any crucial information missing. We respond in more detail also to this comment below. Our responses are highlighted underneath each referee comment using bold text.**

**We thank you for considering our manuscript for publication and look forward to hearing from you soon.**

**Your Sincerely,**

**Allanah Paul**

*On behalf of all co-authors*

**Referee #1 comment: Suggestions for revision or reasons for rejection**

Review of the revision of Paul et al 'Upwelled plankton community modulates surface bloom succession and nutrient availability…… ' submitted to Biogeosciences
General comments.
The results section of this revised version is much more understandable and greatly improved. The authors considered all remarks of the 2 referees and corrected the ms accordingly. To my opinion, the ms is now ready for publication without major modifications.

I have just some minor points: (I refer to the line numbers of the ms with track changes)
Line 292. As NOx is now defined as nitrate + nitrite, the sentence should read : ' Overall, the nitrate + nitrite (NOx)…..'. The same line 393. However, if you want to refer only to nitrate concentrations on lines 292, 293 then you should not write the abbreviation 'NOx' but just write 'nitrate'.
**Thank you for picking up on this. We use $NO_x$ in Line 292 to refer to the measured nitrate + nitrite concentrations (as defined in lines 175 and 290) and high/low nitrate when referring to the nutrient level as we defined in Fig. 1 and have checked for consistency across the manuscript as this was a detail both referees identified.**

Line 394. Remove 'levels'
**This change has been made to line 294, which we assume the referee meant here rather than line 394. After revision, this is now line 292.**

Line 295. Instead of '. .. within each nutrient level…' write ' …with each nutrient treatment (HN, LN)….'
**We thank the referee for the suggestion, but we used the term "within" here deliberately and do not agree that "with" would correctly describe this, however we changed the sentence structure that we hope increases clarity (now lines 293-294). We also kept the use of "level" rather than treatment so this remains consistent with Fig. 1 and the experiment design described in Sect. 2.1.**

From line 299 and after in many places, the authors introduced the term 'incubations' (lines 417, 427, 439, 449, 451, 453 …) or 'levels (line 405). Instead, everywhere, I suggest to write "treatment' as the 6 combinations (HN LN or biology, organic, inorganic) were defined as 'treatments' on line 94
**On line 93, we refer to the "… six treatment combinations … " because in the experimental design, we have 2 nutrient treatments (i.e. "levels") and 3 deep water treatments (what we term "components") that were combined. These are the terms we introduce in Fig. 1 which illustrates the experimental set-up and these treatment combinations and we wanted to have distinct terms for these different factors. We use the term "incubations" to refer to the individual containers e.g. lines 297, 416, 426 including within a nutrient level e.g. line 403. We have reviewed the discussion to ensure there is consistent use of these terms e.g. "incubation" line 339, 343, 347, 426-427 and where possible used "treatment" or "treatment levels" as suggested e.g. lines 329, 340 and hope this clarifies these terms.**

line 416. At the end of the added sentence '….detailed elucidation' I suggest to add something like '…. as information on the heterotrophic community is not available (heterotrophic bacteria, nano and microzooplankton grazers.)'
**This information has been added as suggested to lines 414-415.**

Line 447. Write 'heterotrophic' bacterial production
**This has been added as suggested to line 446.**

Line 502. Write '… were sustained in one of the HN…..
**This has been changed as suggested in line 500.**

Finally, I don't not see in the revised version of the supplement section the evolution with time of

the other groups gated by flow cytometry (picoeuk, micro I, micro II, Crypto, FL4), which was promised in one of the response to the comments of referee 2: ' We will add these figures to the supplementary material to show how these groups varied by time and treatment'
**Thank you for picking up on this oversight. This figure is now available in the revised Supplementary material (Fig. S3) for small and large microphytoplankton, Picoeukaryotes, FL4, Cryptophytes and is referred to in line 313 in the revised manuscript.**

**Referee #2 comment: Suggestions for revision or reasons for rejection**

Although the manuscript has improved substantially, there are two sections in material and methods that I think need further improvement. I am referring to section 2.1 Experimental design and section 2.2 Water collection, incubation setup and sampling; specifically section 2.1 on experimental design.
In my opinion, the modifications introduced in this new version of the manuscript do not help to understand the experimental design. Once screened water and filtered water are clearly differentiated, the whole process becomes clearer.
**This detail is described in Sect 2.2 in the manuscript as is stated in the Figure 1 caption. We also worked to refine again the text in Sect 2.2 to which we hope has improved clarity.**

I do not think it is necessary to name the screened water as unfiltered water in any case, since all collected water was screened to eliminate predators. It is important to make very clear that equal parts of mesocosms water (50% filtered and 50% unfiltered) were used in the inorganic treatments, to which nutrients were added at the level of the deep water in the corresponding HN and LN treatments.
**We agree that this is an important detail and this is indicated in Figure 1 ("Water type added (50%)" and the figure caption ("Sources of inorganic/organic nutrients and the microbial community are in addition to the 50% of mesocosm surface water used as a base in all six treatment combinations."). This is also described in lines 141-144 where we state that 100 L of surface water was added to each tank followed by the addition of 100 L of specific treatment water. We hope this clarifies this important detail sufficiently.**

In the previous version, it was already clear that 50% of filtered water from stations A and B (HN and LN treatments) was added the organic treatments.

Honestly, I think this part of the manuscript needs improvement. The experiment is complex and the reader must easily understand what has been made, in order to also be able to follow and understand the results.

**We agree that it is a complex experimental set-up and found it challenging to present this in an easily digestible form.  As far as we understand, the feedback from the Referee is related to the order of the presented material rather than any missing details. The current figure placement inside the manuscript file, which will likely change in any published manuscript due to formatting, may add to confusion here. We followed what we thought was a logical order: Section 2.1 describes the experimental design and the distinction between the six treatment combinations used to address our research questions. Section 2.2 describes the water collection in detail, starting with the geographic location and sampling procedures for both the subsurface water and following the work flow steps including the filtration steps and preparation of the six treatment combinations.**

Other minor things that I think need correction.
Line 122. Only 40 L of water were collected in the mesocosms?

**400 L of water was collected in total. This has been corrected.**

Line 156. Light does not pass through black tubes. In the original version black tubes were not mentioned.
**Yes, the incubator tubs were made of black plastic but have open tops that allowed the light to penetrate. The measured lux values were taken from inside the incubators and are representative measurements of the light conditions the containers received in these tubs. If that extra detail is confusing, it can be removed.**

Line 292. Initial nitrate (NOx)… I understand that it should be initial inorganic nitrogen. At least this is what can be read in other parts of the text. The entire text should be revised to correct these uncertainties.
**Thank you for this suggestion. Line 290 (previously 292) has been corrected to read "… initial nitrate + nitrite ($NO_x$) …". A similar change was made to line 173 to clearly state this definition. We revised the entire text to clarify this. There are still some cases where "inorganic nitrogen" is used (e.g. lines 115, 125) but these are not related to measured NOx concentrations or nitrate levels in this study.**

Lines 293-294. This sentence seems somewhat confusing to me, because it can be understood that initial concentrations of inorganic nitrogen were similar in HN and LN. Perhaps something more explicit like: Initial NOx concentrations were similar in all HN treatments ( 7.72…) and all LN treatments (2.56…)
**We have modified this sentence (now lines 292-293) which we hope more clearly defines the comparison we want to make here between the two nitrate levels: " Initial $NO_x$ concentrations were similar in all 12 high nitrate (HN, $[NO_x]$ = 7.72 ± 0.46 µmol $L^{-1}$, mean ± s.d., n = 12) and in all 12 low nitrate treatments (LN, $[NO_x]$ = 2.56 ± 0.54 µmol $L^{-1}$ mean ± s.d., n = 12, Fig. 3B)".**

Line 341. Delete the second were from … were also were observed…
**This has been corrected.**

Line 405. While nitrate concentrations… should read while inorganic nitrogen concentrations…
**This has been changed to $NO_x$ to more precisely indicate the variable we measured and are referring to here.**

Line 528. Such physical factors… should read other physical factors.
**We do not consider this change is necessary because in this sentence we are indicating specifically which physical factors we referred to as "other physical factors" in line 527.**

Lines 540-542. Here it is assumed that they are diatoms. I it should be conditional: probably diatoms, as silicate reduction suggests.

**We have modified lines 538/540 to address this comment and use "silicifying phytoplankton" instead of "diatoms" in other cases in the revised manuscript.**